# Prevalence, spatial patterns and determinants of zero-dose vaccination among children aged 12–23 months in Sub-Saharan Africa: A multilevel spatial analysis

**Tadesse Tarik Tamir**[1]*, **Muluken Chanie Agimas**[2], **Dessie Abebaw Angaw**[2]

**1** Department of Pediatrics and Child Health Nursing, School of Nursing, College of Medicine and Health Sciences, University of Gondar, Gondar, Ethiopia, **2** Department of Epidemiology and Biostatistics, Institute of Public Health, College of Medicine and Health Sciences, University of Gondar, Gondar, Ethiopia

* tadestar140@gmail.com

## Abstract

### Introduction

Vaccination is one of the most cost-effective strategies for enhancing public health and well-being. Despite widespread immunization efforts, zero-dose vaccination among children remains a pressing public health challenge in sub-Saharan Africa (SSA) and there exists a notable gap in research regarding its current burden, spatial distribution and determinants. Hence, this study was aimed at assessing prevalence, spatial patterns and determinants of zero-dose vaccination among children aged 12–23 months in SSA, 2025.

### Methods

A secondary data analysis of community-based cross-sectional Demographic and Health Survey data from 2015 to 2024 was conducted using a total weighted sample of 58,443 children aged 12–23 months. Descriptive statistics were summarized using texts, tables, and figures, and a multilevel spatial analysis was applied to identify the determinants of zero-dose vaccination. Adjusted odds ratios (AORs) with 95% confidence intervals (CIs) were computed to determine the level of significance, with a p-value < 0.05 considered statistically significant.

### Results

The prevalence of zero-dose vaccination among children aged 12–23 months in SSA was 13.3% with 95% CI (10.2%, 16.3%). The hot spot clusters were located in most of Nigeria, Angola, Ethiopia, Madagascar, Guinea, Cote d'Ivoire, the southern border of Mauritania, and southeastern of Mali and Mozambique. The poorest household wealth index (AOR = 1.30; 95% CI: 1.10–1.53), home delivery (AOR = 2.60;

**Data availability statement:** All relevant data are within the paper and its Supporting Information files.

**Funding:** The author(s) received no specific funding for this work.

**Competing interests:** The authors have declared that no competing interests exist.

95% CI: 2.39–2.83), having no media exposure (AOR = 1.25; 95% CI: 1.14–1.35), no (AOR = 2.41; 95% CI: 2.17–2.68) or low antenatal care visits (AOR = 1.37; 95% CI: 1.26–1.49), having no tetanus toxoid vaccination (AOR = 1.63; 95% CI: 1.49–1.79), having no (AOR = 1.81; 95% CI: 1.62–2.03) or only primary maternal education (AOR = 1.28; 95% CI: 1.16–1.43), perceived problem of distance to health facility (AOR = 1.13; 95% CI: 1.05–1.21) and rural residence (AOR = 1.25; 95% CI: 1.07–1.67) were significantly associated with zero-dose vaccination.

## Conclusion

In SSA, zero-dose vaccintion was high compared to the global target of no tolerance for zero-dose. The distribution of zero-dose vaccination was spatially clustered with individual and community level factors significantly associated with it. Therefore, spatially targeted interventions taking these factors into account should be implemented by stakeholders in the SSA region to address zero-dose children.

## Introduction

Vaccination is defined as a simple, safe, and effective method of protecting individuals against harmful diseases prior to actual exposure [1,2]. It works by using the body's natural defenses to build resistance to specific infections, thereby strengthening the immune system [1]. A vaccine prepares and empowers the body so that it may generate antibodies against certain pathogens, viruses, or bacteria without permitting the organism to damage the body [1]. This enables the body's defenses to 'remember' the particular illness which they may encounter later on, and respond more proficiently [3]. Various methods of administering vaccine include injections, oral pills, ointments, or nasal sprays. They are essential for averting grave medical problems and have the ability to shield individuals from the early stage of infant life to old age [2]. Getting vaccinated ensures protection to the individual, which also strengthens community immunity, ensuring safety for those who are unable to get the vaccine, including newborns or those with some medical issues [3]. Preventive health care starts with vaccination which is not only a necessity for children but also a legal condition for social involvement [4,5]. Internationally, it is acclaimed for being one of the most economically efficient measures intended to boost public health and foster wellbeing at the same time catalyzing sustainable development [6]. Vaccination programs are foreseen to avert about 5.1 million deaths yearly from diseases that have available vaccines [7]. Currently, over 100 million children are vaccinated against diphtheria, tetanus, whooping cough, tuberculosis, polio, measles, and Hepatitis B [8]. Immunization has proven to be a reliable public health measure; however, some children are still not vaccinated with the first dose of diphtheria- tetanus- pertussis vaccine known as zero-dose children [9,10].

Globally, 13.8 million children were zero-dose in 2019 [10] and by 2021 this figure rose to 18.2 million, which represents more than 70% of unvaccinated children aged 12 years and younger [11,12]. This increase was even more pronounced during the

covid-19 pandemic when childhood vaccinations plummeted and millions became susceptible to easily preventable diseases [13]. Africa remains one of the regions with a disproportionately high burden of the zero-dose population [10] with unfathomable surge of 6.2 million to over 7.7 million unvaccinated children aged 12–23 months between 2019 and 2022 [14]. To correct these disparities, global health stakeholders have put in place various strategies and interventions. The WHO has developed the Global Routine Immunization Strategies and Practices (GRISP) framework, and it contains nine transformational investments that have priority [15]. These include (1) investing in a capable national immunization team with adequate authority and resources, (2) tailoring strategies to reach underserved and zero-dose populations, (3) establishing a coherent planning cycle that integrates strategic and operational plans, (4) ensuring adequate and timely funding at the operational level, (5) building the capacity and performance of vaccinators and district managers, (6) modernizing vaccine supply chains to ensure timely and equitable distribution, (7) enhancing information systems to track individual vaccination status, (8) promoting life-course vaccination beyond infancy, and (9) fostering shared responsibility with communities to generate demand and ensure service quality [15]. GRISP also supports a life-course perspective of immunization, acknowledging the necessity of long-term coverage in early childhood. In Supporting GRISP, the Global Vaccine Action Plan (GVAP) and its revision, the Immunization Agenda 2030 (IA2030), support universal vaccination through integrated, people-oriented delivery systems [16]. In low-and middle-income countries, and especially in sub-Saharan Africa, other interventions have also shown promise. These involve community outreach, home visits, integration of immunization within other health services, and involvement of local leaders [17]. However, these strategies need up-to-date evidence on magnitude, geographic location and contributing factors of the problem.

Earlier observational studies in LMICs including sub-Saharan African countries have identified various factors influencing childhood immunization rates. Sociodemographic factors include maternal age, parental education, marital status, occupation, family income, wealth index, vaccination knowledge, and ethnicity [18–22]. Additional factors include geographic location, the availability of healthcare facilities, community awareness and attitudes regarding vaccinations, and access to healthcare services [18,23,24]. Obstetric factors are also important, such as the location of delivery, the utilization of prenatal and postnatal care, and the time between prior births [22,25–27]. However, to the best of researchers' knowledge, no previouse studies have examined the spatially adjusted individual and community level determinants of zero-dose vaccination.

Zero-dose children lack access to basic healthcare and face interconnected challenges, and they are often isolated in urban slums or conflict-affected rural areas, account for one-third of all child deaths in low- and middle-income countries [11]. Reaching them is vital to reaffirming the global pledge of "leaving no one behind" [28], yet sub-Saharan Africa—a region grappling with strained health systems and socioeconomic inequities—remains understudied and reported interms of zero-dose vaccination. Despite decades of global investment on childhood vaccination, a substantial number of children remain zero-dose. This exclusion not only puts their lives at risk but also undermines progress of immunization agenda 2030 to toward closing immunization gaps and leave no child behind [29]. Sub-Saharan Africa is one of the regions with the highest global burden of zero-dose vaccination among children [14]. This is largely due to a complex web of several factors. Despite the critical nature of the problem and need of frequent evaluation; there is dearth of research on current prevalence, regional distribution and its determinants especially using current definition of zero-dose and the spatial hierarchical modeling approach.

This study addresses these gaps by applying a spatial multilevel modeling approach to examine determinants of zero-dose vaccination with its prevalence and spatial distribution among children aged 12–23 months in Sub-Saharan Africa. By using this method, we aimed to uncover not just where zero-dose children are concentrated, but also why they are being left behind—accounting for both individual and community-level factors with consideration of spatial correlation. This methodological approach fills a crucial gap in the literature, where spatial and hierarchical dimensions are often studied in isolation or not at all. Therefore, the significance of this study lies in its potential to reveal overall prevalence, identify high-prevalence countries, high-risk clusters, and spatially adjusted factors contributing to zero-dose vaccination

 

in Sub-Saharan African region using nationally representative survey data. These findings would inform more precise and equitable health planning. Rather than relying on one-size-fits-all strategies, our findings can help policymakers and health planners design interventions that are tailored to the unique challenges of high-risk areas. This is especially important in resource-limited settings, where efficient allocation of funds and services can make the difference between inclusion and neglect.

## Methods and materials

### Study design and period

A secondary data analysis of community-based cross-sectional DHSs from 2015 to 2024 was conducted, with the data extraction and analysis period from April 1 to May 30, 2025.

### Study setting

Sub-Saharan Africa, a geographically expansive region located south of the Sahara Desert, encompasses Central Africa, East Africa, Southern Africa, and West Africa [30]. This region is home to diverse populations, each experiencing varying levels of access to healthcare services and environmental conditions that significantly impact health outcomes [31]. For this study, a total of twenty-eight sub-Saharan African countries were selected from the four regions of Africa: Eastern Africa (Burundi, Ethiopia, Kenya, Madagascar, Malawi, Mozambique, Rwanda, Tanzania, Uganda, Zambia, Zimbabwe), Central Africa (Angola, Cameroon, Gabon), Western Africa (Benin, Burkina Faso, Côte d'Ivoire, Gambia, Ghana, Guinea, Liberia, Mali, Mauritania, Nigeria, Senegal, Sierra Leone), and Southern Africa (Lesotho, South Africa), as shown in Fig 1. The selection of these countries was based on predefined inclusion criteria.

### Data and source

The study used data extracted from the DHS datasets conducted from 2015 to 2024. The dataset was accessed from the Monitoring and Evaluation to Assess and Use Results Demographic and Health Survey (MEASURE DHS) program official website (http://www.dhsprogram.com) through registration and request. The only prerequisite for accessing the dataset is registration. The children recode (KR) data, which was extracted from the DHS dataset, was used in this study.

### Populations

**Source population.** All children aged 12–23 months old in sub-Saharan Africa were taken as a source population.

**Study population.** All children aged 12–23 months residing in the enumeration areas where DHS were conducted from 2015 to 2024 in sub-Saharan African countries, and who were available during data collection, were included as the study population.

### Eligibility criteria

**Inclusion criteria.** All children aged 12–23 months during the survey period, residing in selected enumeration areas of sub-Saharan African countries with DHSs conducted from 2015 to 2024, were included in the study (Fig 2).

**Exclusion criteria.** Children aged 12–23 months in sub-Saharan African countries where data on routine vaccination is unavailable in the DHSs from 2015 to 2024 were excluded from this study. Additionally, enumeration areas lacking coordinates were excluded from the spatial analysis.

### Sample size determination and sampling procedures

The DHS employs a stratified two-stage cluster sampling technique [32]. The sample is stratified by geographic region and urban/rural areas within each region. This process involves two stages: first, the selection of enumeration areas

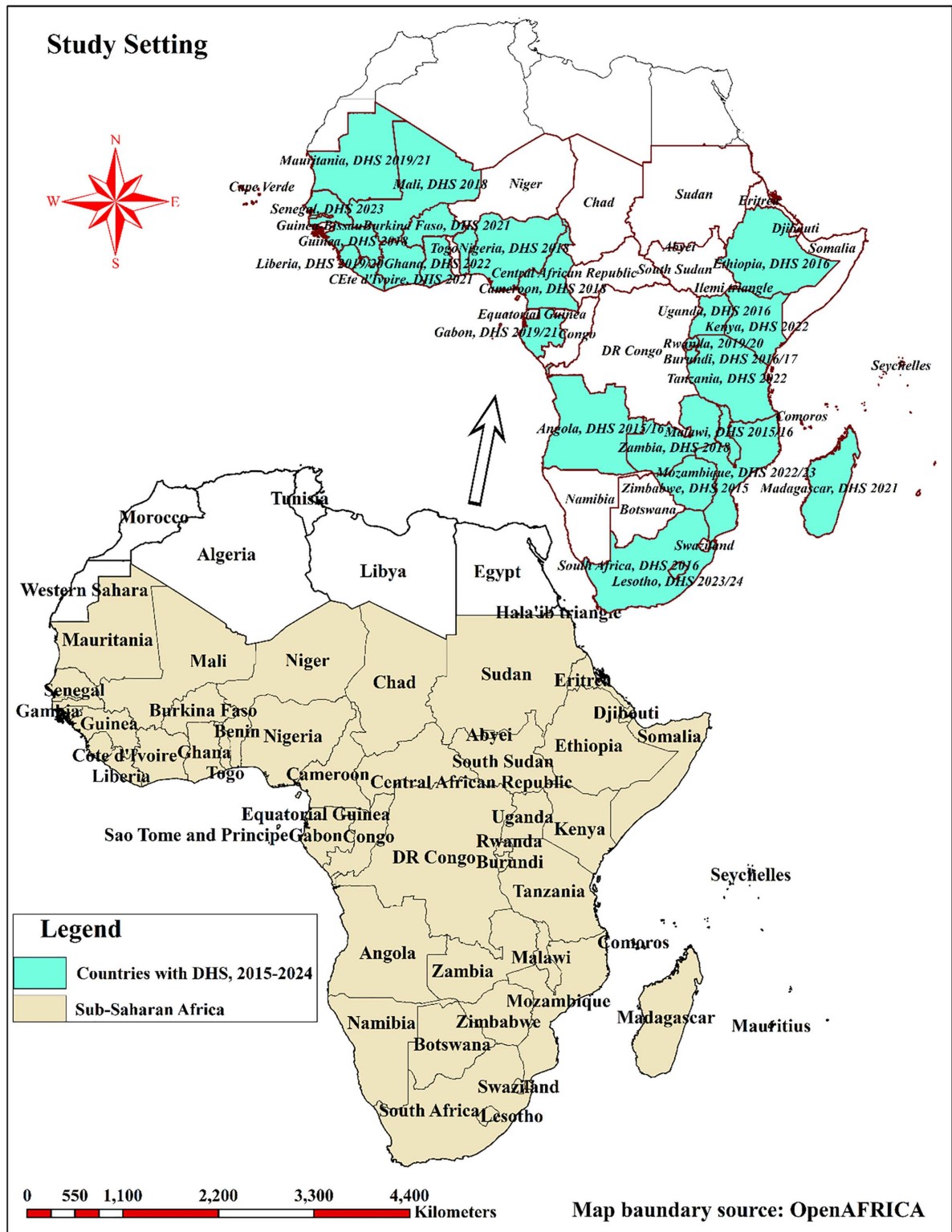

**Fig 1. Graphic description of study setting for prevalence, spatial disparity and determinants of zero-dose vaccination among children aged 12-23 months in sub-Saharan Africa; Map created using Africa shapefiles – Admin Level 0 under Open Data Commons Open Database License from openAFRICA.**

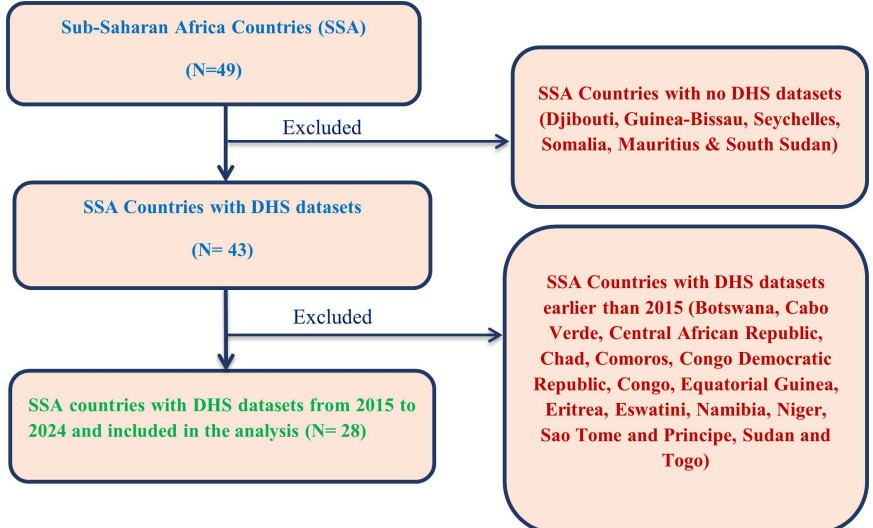

**Fig 2. Eligibility criteria of prevalence, spatial distribution and determinants of Zero-dose vaccination among children aged 12-23 months in sub-Saharan Africa.**

(sampling clusters), and second, the drawing of a sample of households within each enumeration area [33]. In this study, a total weighted sample of 58,443 of children aged 12–23 months from all 28 countries were used for analysis (Table 1).

## Variables of the study

**Dependent variables.** The outcome variable of the study was zero-dose status. This variable was categorized as a binary measure (yes/no) with a value of 1 that signifies the child was yes for zero-dose and a value of 0 indicates that the child was no for zero-dose.

**Independent variables.** The independent variables of the study were grouped int to two levels: individual and community. At the individual level, the factors considered include socio-demographic and economic characteristics, parental factors, and obstetric-related factors. Socio-demographic and economic characteristics encompassed the child gender, household head gender, the household wealth index, the number of children in the household, media exposure and distance to health facilities. Parental factors included maternal age, education, occupation, and marital status, as well as paternal education and occupation. Obstetric-related factors involved the place of delivery, delivery assistance, the number of antenatal care visits, tetanus toxoid vaccination and birth order. Community-level factors included the place of residence and sub-regions of Africa in sub-Sahara. By examining these variables at both the individual and community levels, we aimed to gain a comprehensive understanding of the factors influencing zero-vaccination rates (Fig 3).

## Operational definitions

**Zero-dose Children:** According to the World Health Organization (WHO) and the Immunization Agenda 2030 (IA 2030), zero-dose children are defined as those who have not received any doses of the diphtheria-tetanus-pertussis vaccine by the age of 12 months [16,45]. This definition serves as an important measure of immunization access, highlighting the significant barriers these children face in receiving essential vaccinations. The identification of zero-dose children is operationally measured by determining those who have not received their first dose of DTP, which acts as a proxy for overall immunization coverage and access to healthcare services. The rationale for this classification is underscored by IA 2030's goal to ensure that no child is left behind in vaccination efforts, emphasizing the need for targeted interventions to reach

**Table 1. Sample size determination for prevalence, spatial disparity and determinants of zero-dose vaccination among children aged 12-23 months in sub-Saharan Africa.**

| Country | Survey Year | Unweighted Sample Size | | Weighted Sample Size | |
|---|---|---|---|---|---|
| | | Frequency (n) | Percent (%) | Frequency (n) | Percent (%) |
| Angola | 2015/16 | 2,845 | 4.81 | 2595 | 4.44 |
| Benin | 2017/18 | 2,522 | 4.26 | 2515 | 4.30 |
| Burkina Faso | 2021 | 2,313 | 3.91 | 2299 | 3.93 |
| Burundi | 2016/17 | 2,596 | 4.38 | 2681 | 4.59 |
| Cameroon | 2018 | 1,824 | 3.08 | 1900 | 3.25 |
| Cote d'Ivoire | 2021 | 1,920 | 3.24 | 1819 | 3.11 |
| Ethiopia | 2016 | 1,929 | 3.26 | 2004 | 3.43 |
| Gabon | 2019/21 | 1,271 | 2.15 | 1289 | 2.21 |
| Gambia | 2019/20 | 1,582 | 2.67 | 1456 | 2.49 |
| Ghana | 2022 | 1,973 | 3.33 | 1823 | 3.12 |
| Guinea | 2018 | 1,408 | 2.38 | 1384 | 2.37 |
| Kenya | 2022 | 3,679 | 6.21 | 3324 | 5.69 |
| Lesotho | 2023/24 | 529 | 0.89 | 490 | 0.84 |
| Liberia | 2019/20 | 1,063 | 1.80 | 937 | 1.60 |
| Madagascar | 2021 | 2,345 | 3.96 | 2337 | 4.00 |
| Malawi | 2015/16 | 3,248 | 5.49 | 3230 | 5.53 |
| Mali | 2018 | 1,946 | 3.29 | 2048 | 3.50 |
| Mauritania | 2019/21 | 2,119 | 3.58 | 2120 | 3.63 |
| Mozambique | 2022/23 | 1,727 | 2.92 | 1807 | 3.09 |
| Nigeria | 2018 | 6,059 | 10.23 | 6143 | 10.51 |
| Rwanda | 2019/20 | 1,572 | 2.66 | 1633 | 2.79 |
| Senegal | 2023 | 2,057 | 3.47 | 1949 | 3.33 |
| Sierra Leone | 2019 | 1,861 | 3.14 | 1838 | 3.14 |
| South Africa | 2016 | 670 | 1.13 | 677 | 1.16 |
| Tanzania | 2022 | 2,143 | 3.62 | 2180 | 3.73 |
| Uganda | 2016 | 2,922 | 4.94 | 2859 | 4.89 |
| Zambia | 2018 | 1,928 | 3.26 | 1891 | 3.24 |
| Zimbabwe | 2015 | 1,151 | 1.94 | 1216 | 2.09 |
| **Total** | **2015-2024** | **59,202** | **100.00** | **58,443** | **100.00** |

underserved populations [45]. In this study, zero-dose status was determined through two primary methods employed in DHSs: direct verification of vaccination cards when available and maternal self-reporting in cases where the card is absent. This approach ensures a comprehensive assessment of vaccination status, capturing both documented and self-reported data.

**Household Wealth Index:** In DHS, households are graded based on the number and type of consumer goods they own, which can range from a television to a bicycle or automobile, as well as housing qualities such as drinking water sources, bathroom facilities, and flooring materials. Principal component analysis is used to calculate these scores. National wealth quintiles are calculated by assigning the household score to each usual household member, ranking each person in the household population based on her or his score, and then dividing the distribution into five equal groups (poorer, poorest, middle, richer and rich), each of which represents 20% of the population [32].

**Media Exposure:** This variable was derived from three potential media sources: watching television, listening to the radio, and reading magazines or books. Mothers who were exposed to at least one of these media sources were

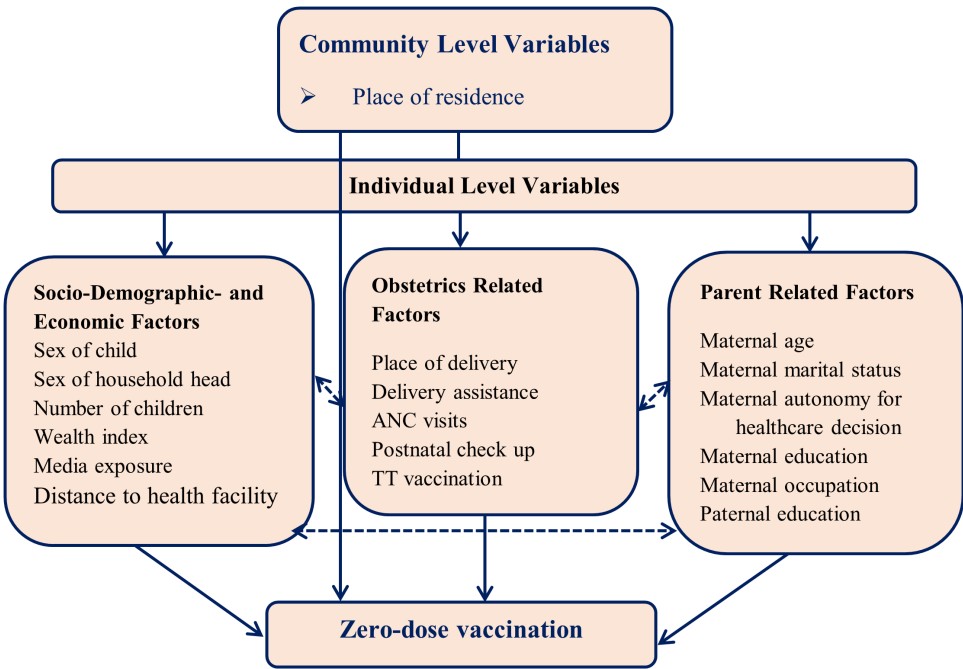

**Fig 3. Conceptual framework for associated factors of zero-dose vaccination among children in sub-Saharan Africa accessed from literature [18–22,34–44].**

classified as having mass media exposure. Conversely, those who were not exposed to any of these three media sources were classified as having no mass media exposure [46,47].

## Data collection tool and procedure

The DHS utilizes a standardized model questionnaire designed to ensure consistency and comparability of data across countries. Each questionnaire is accompanied by a detailed rationale explaining the inclusion of specific questions or sections. The model questionnaire undergoes rigorous review and refinement during each phase of the DHS process. Countries are encouraged to adopt the complete model questionnaire to facilitate cross-country data comparability. However, they have the flexibility to add questions that address particular national interests. Conversely, questions deemed irrelevant to a specific context may be omitted. Data is collected through structured interviews with eligible participants from selected households. During the survey sampling process, cluster coordinates are recorded using a GPS receiver [48].

## Data quality control

Ensuring high data quality is a fundamental aspect of the DHS Program [48]. It is accomplished by pretesting questionnaires, data collector training, and following standardized procedures for handling missing or inconsistent data. In the lack of vaccination cards, maternal report is utilized to determine if a vaccine has been administered or not; however, in case certain vaccines are not found on the card, maternal report is utilized, and their lack leads to the assumption that the vaccine was not given [49]. Coding and recoding procedures adhere to strict guidelines to preserve data integrity. Special codes are assigned during secondary editing to flag implausible values when corrections are unfeasible, ensuring analyses avoid over- or underestimation of outcomes [48].

## Data management and analysis

Data cleaning and descriptive statistical analysis specifically characteristics of study subjects, chi-squared association and prevalence were performed using Stata 17. The data preparation for descriptive spatial analysis was done using Microsoft Excel 2021 and actual analysis was performed using ArcGIS 10.8 and SaTScan 10.2.3. Importantly, R 4.5.0 was used for spatial multilevel analysis in R studio. The dataset was cleaned, organized, and georeferenced by linking geographic coordinates to shapefiles for spatial representation. Sampling weights were applied using the svyset and svy commands to account for the complex survey design of the DHS data. This involved utilizing weighting variables, including the sampling weight (v005), primary sampling unit (v021), and strata for sampling design (v023). This approach ensures that the sample is representative of the broader population. By incorporating these sampling weights, we aimed to adjust for the unequal probabilities of selection inherent in the survey design, thereby enhancing the accuracy and validity of our findings and allowing for the drawing of valid conclusions. These values were calculated from the DHS children recode dataset. Descriptive statistics were used to summarize population characteristics, with subgroup analyses comparing for source of heterogeneity in our pooled estimate of zero-dose prevalence. The distribution of zero-dose children was visualized through tables, forest plots and maps to elucidate spatial and demographic patterns. Variance inflation factor was computed for any multicollinearity between independent variables of the study. Additional details about the data management and analysis are available in our S1 File.

**Spatial analysis.** Spatial data refers to information about geographical locations, and spatial data analysis involves examining this data while considering the spatial locations. A key aspect of spatial statistical models is that values of attributes that are close to each other are more statistically dependent than those that are farther apart, a concept known as the First Law of Geography [50]. Spatial dependence happens when values at one location are influenced by neighboring observations, necessitating unique analysis methods. The proximity between areas affects relationships, a phenomenon called spatial autocorrelation, which impacts spatial prediction techniques based on regionalized variable theory [51].

**Spatial autocorrelation.** Spatial autocorrelation measures how similar or dissimilar values are at nearby locations. A useful statistic for measuring spatial autocorrelation is Moran's I, which standardizes the spatial auto-covariance. This study used Global Moran's I to determine spatial dependency and clustering of zero-dose vaccination accross sub-Saharan Africa. Global spatial autocorrelation is a statistical measure that evaluates the overall degree of spatial dependency or clustering of a variable across an entire study area. It quantifies whether similar values (high or low) of a spatial variable tend to occur near each other more than would be expected under a random spatial distribution [52–54].

**Hotspot analysis.** This study employed optimized hotspot analysis (OHA) with the Getis-Ord Gi statistic* to identify statistically significant spatial clusters of zero-dose vaccination among children that advances traditional hotspot techniques. OHA automates the selection of critical spatial parameters, such as neighborhood scales and distance thresholds, through iterative optimization algorithms, thereby minimizing subjectivity inherent in conventional hotspot mapping [55]. The Gi* statistic evaluates spatial associations by computing normalized z-scores and p-values for each location, distinguishing clusters where the concentration of zero-dose children is significantly higher (hotspots) or lower (coldspots) than expected under spatial randomness [55]. Importantly, OHA incorporates False Discovery Rate (FDR) correction, a statistical adjustment that limits the proportion of false-positive clusters identified due to multiple hypothesis testing across numerous spatial units. By controlling the FDR at a predefined threshold, the analysis ensures that only regions with robust evidence of clustering—rather than artifacts of random noise—are flagged for intervention, enhancing the reliability of results in large-scale spatial datasets [29].

**Spatial scan statistical analysis.** Spatial clustering patterns of zero-dose vaccination was analyzed using SaTScan software version 10.2.3, a spatial scan statistic designed to detect significant disease or intervention clusters while adjusting for underlying population heterogeneity [56]. The Bernoulli probability model was applied to case-control data, where cases represented unvaccinated children and controls comprised vaccinated children, georeferenced to their

residential coordinates [56]. A dynamic circular scanning window, whose radius expanded from zero to a maximum threshold (50% of the population at risk), systematically traversed the study area to identify candidate clusters. For each potential cluster, the method computed a log-likelihood ratio (LLR) to quantify the probability of excess cases within the window compared to outside it, a relative risk (RR) to estimate the likelihood of being unvaccinated inside versus outside the cluster, and a Monte Carlo-derived p-value to assess statistical significance (P = 0.05) [56]. The outputs of SaTScan analysis were mapped using ArcGIS 10.8 to graphically depict the cluster windows.

**Spatial interpolation.** Sampling does not allow for the collection of every measurement from all locations, so values for unknown data at specific sites are gathered. These samples are then utilized to estimate values at unmeasured points. Factors such as size, orientation, shape, and the spatial arrangement of the measured points influence the accuracy of these predictions. The process of estimating unknown values based on known sample points is known as interpolation. Interpolation refers to the estimation of attribute values at locations that have not been sampled, using measurements from other points within the same region [57]. This technique is primarily employed to transform data from a limited number of observations into continuous fields, enabling analysis of the spatial patterns of other entities based on the sampled data. Consequently, spatial interpolation serves as a method for deriving surface data from point observations, using known values to estimate those of unknown points that make up the surface [57]. This approach is grounded in Tobler's first law of geography, which states, 'Everything in space is interconnected, but nearby points are more likely to exhibit similarities than those that are farther apart [58].

Kriging is a prominent method for geostatistical interpolation that has been utilized [57]. It is an estimation technique that leverages known values along with the semivariogram to determine unknown values. Named after South African statistician D.G. Krige, kriging incorporates measures of error and uncertainty in its estimations. 'the method relies on a random field and several assumptions, such as stationarity and spatial ergodicity, which simplify the required information into a variogram that can be derived from available measurements [59].' Using the semivariogram, optimal weights are assigned to known values to estimate unknown ones, with the variogram varying based on distance and the weights influenced by the distribution of known samples. Kriging operates under the premise that spatial variation is neither entirely random nor fully deterministic, focusing on regionalized variables. Statistically, the validity of the variogram may conflict with the measured values. Generally, as the study area increases and more data is collected, the variogram becomes more reliable. Conversely, the validity of local analyses tends to be lower because there is less data available. A general equation for kriging is $Z(s) = \mu(s) + \varepsilon'(s)$ [57]. Where $Z(s)$ represents the variable of interest at location 's', $\mu(s)$ is the deterministic trend component, which can be thought of as the expected value or mean of $Z(s)$ and represents the random, autocorrelated errors, capturing the spatial correlation between observations. The location '**s**' typically indicates the x and y coordinates in a spatial context.

We utilized an ordinary kriging (OK) technique for interpolation after comparing it with other techniques (see S1 File). OK is a widely used geostatistical method for spatial interpolation, represented by the equation $Z(s) = \mu + \varepsilon'(s)$, where $Z(s)$ is the variable of interest at location $S$, μ is an unknown constant mean, and $\varepsilon'(s)$ represents the spatially correlated stochastic part of the variation. One of the key advantages of ordinary kriging is that it is considered the Best Linear Unbiased Estimator (BLUE), meaning it provides the most accurate and unbiased predictions based on the available data [60]. This method leverages the semivariogram to account for spatial autocorrelation, making it particularly effective in capturing the spatial structure of the data. Unlike simple kriging, which requires a known mean, ordinary kriging does not necessitate prior knowledge of the mean, making it more flexible and applicable in various scenarios where the mean is unknown or variable [60].

## The regression model

A multilevel model is a statistical technique that extends ordinary regression analysis to situations where the data are multilevel or hierarchical, correcting for biases in parameter estimates resulting from clustering [61,62]. When observations

are nested into higher-level units, they are no longer independent [62]. A spatial multilevel model is further extension of multilevel model to a model equation which accounts spatial correlation using spatially structured random effect [63].

To address the hierarchical structure of the DHS data, where individuals (Level 1) are nested within geographically defined clusters or communities (Level 2)—and to account for spatial autocorrelation in zero-dose prevalence due to unmeasured environmental, health system, or socio-cultural factors, we employed a two-level multilevel spatial logistic regression model. This model was implemented in R 4.5.0 using the sdmTMB package, which extends the generalized linear mixed model (GLMM) framework by incorporating a spatial random effect at the cluster level. This spatial effect is modeled as a Gaussian Markov Random Field (GMRF), capturing residual spatial dependence after adjusting for measured covariates. This approach mitigates bias and information loss that can occur in non-spatial models.

The GMRF was constructed using the stochastic partial differential equation (SPDE) approach with the Matérn covariance function. This method leverages the computational efficiency of GMRFs—enabled by sparse precision matrices that reflect local dependencies—and the flexibility of the Matérn function to model spatial range and variance [64].

We set the smoothness parameter $\nu = 0$ (equivalent to $\alpha = 1$ in the SPDE formulation for 2D space), corresponding to an exponential covariance structure, which is a common and computationally efficient default. The spatial domain was discretized using a finite element mesh, constructed via k-means clustering with 5,000 knots over a domain defined by the convex hull of cluster locations, buffered by 10% of the maximum inter-cluster distance [65].

The GMRF, solved using a Gaussian white noise-driven stochastic partial differential equation (SPDE):

$$W(s) = (\kappa^2 - \Delta)^{\alpha/2} x(s) \text{ [65]}.$$

where W(s): Represents Gaussian white noise with a mean of 0 and unit variance.

$\kappa$: A parameter that influences the spatial scale or range of the process.

$\Delta$: The Laplace operator, which is a differential operator given by the sum of second partial derivatives.

$\alpha$: A positive parameter that affects the smoothness of the process and is greater than 0.

x(s): The spatial process being modeled.

Four spatial multilevel models were fitted: a null model (Model 0), an individual-level factors model (Model I), a community-level factors model (Model II), and a full model including both levels (Model III). The best-fitting model was selected based on the lowest Akaike Information Criterion (AIC), Bayesian Information Criterion (BIC) and maximum likelihood criterion at convergence. Parameters (fixed effects, non-spatial random effects, and Matérn parameters) were estimated via maximum likelihood using Template Model Builder (TMB) within sdmTMB.

Factors with p value of less than 0.25 in the bivariate analysis were included in multivariable analysis. Adjusted odds ratios with 95% confidence intervals, derived from fixed effect coefficients and standard errors accounting for all random effects, was presented for interpretation, with statistical significance declared at $p < 0.05$.

## Random effect

To account for unobserved heterogeneity and spatial dependence in the data, the spatial multilevel logistic regression model incorporated both unstructured and structured random effects. Unstructured random intercepts at the community level captured residual variability between clusters not explained by observed covariates, while a structured spatial random field modeled geographic clustering. The contribution of the unstructured random effect to total variability was quantified using the intraclass correlation coefficient (ICC), calculated as ICC = $\frac{VC}{VC+3.29} \times 100\%$, where $V_C$ denotes the between cluster variance and ($3.29 = \pi^2/3$ represents the variance of the standard logistic distribution. To interpret the magnitude of unexplained heterogeneity between clusters on the odds ratio scale, the median odds ratio (MOR) was computed using the formula, MOR = $e^{0.95\sqrt{VC}}$, which reflects the median increase in odds when comparing two individuals with identical covariates from different clusters. Additionally, the proportional change in variance (PCV) was used to assess the explanatory power of covariates, defined as PCV = $\frac{Vnull-Vc}{Vnull} \times 100\%$; where $V_{null}$ is the between-cluster variance in the null model and $V_C$ is the variance in the adjusted model.

The spatial dependence structure in the model was characterized using the Matérn covariance function, a flexible and widely used function in spatial statistics that allows for control over both the range and smoothness of spatial correlation. The Matérn covariance function is defined as:

$$c_{\sigma^2, \nu, \kappa}(h) = \frac{\sigma^2}{[2\nu - 1\Gamma(\nu)]} (\kappa\|h\|) \nu K\nu (\kappa\|h\|)$$

Where $\sigma^2$: Variance parameter, which scales the overall magnitude of the covariance.

$\nu$: Shape parameter, defined as $\nu = \alpha - d/2 > 0$, where d is the dimension of the space.

$\kappa$: Controls the range of the spatial correlation. Larger values of $\kappa$ correspond to shorter-range dependencies.

$\|h\|$: The Euclidean distance between two points.

$K\nu$: The modified Bessel function of the second kind, which is used to model the decay of the covariance with distance.

$\Gamma(\nu)$: The Gamma function, which generalizes the factorial function to non-integer values [64].

## Ethical consideration

The study was conducted after obtaining a permission letter from the DHS Program through an online request to access DHS data from Sub-Saharan African countries, following the review of submitted brief descriptions of the survey. To protect the confidentiality of survey respondents, spatial analysis was limited to aggregated data, ensuring that no individual data points can be identified, with ethical acceptability defined as 5 km for rural areas and 2 km for urban areas. The datasets were treated with the utmost confidentiality, and issues related to informed consent, anonymity, and privacy was ethically handled by the DHS office. We did not manipulate or apply the microdata beyond the scope of this study, and there was no patient or public involvement in this research.

## Results

### Characteristics of study subjects

This study analyzed zero-dose vaccination rates among 58,443 children aged 12–23 months across sub-Saharan Africa, with a near-equal gender distribution (50.84% male, 49.16% female). More than one-third [21,470 (36.75%)] of children were members of households which had four or more living children in the households and 13,289 (22.74%) of children were from household with poorest wealth index. Notably, two-third (65.13%) of children were born to mothers with no media exposure. Importantly, the majority (73.73%) of children were born at health facilities and 44,280 (75.02%) were born with a skilled birth attendant. Moreover, the distribution of children interms of residence was 20,122 (34.43%) for urban and 38,322 (65.57) for rural.

At the individual-level, zero-dose vaccination did not significantly vary by child gender, with comparable rates among males (14.57%) and females (14.85%) ($\chi 2 = 1.14$, $p = 0.285$). However, household gender dynamics played a role: male-headed households reported a higher zero-dose rate (15.34%) compared with female-headed households (12.37%), a difference that was confirmed by statistical figures ($\chi 2 = 38.24$, $p < 0.001$). The zero-dose status was varied with the index of household wealth. Children from the poorest households faced a zero-dose rate of 22.91%, nearly three times higher than those from the richest households (7.09%). Notably, children of mothers with no formal education had a 25.34% zero-dose rate, compared to 7.56% for mothers with secondary or higher education. Similarly, zero-dose rates were lower (13.31%) for employed mothers compared to 16.86% for unemployed mothers). In terms of antenatal care, children of mothers who attended no ANC visits had a zero-dose rate of 40.14%, dropping to 13.99% for 1–3 visits and 8.48% for four or more visits. At the community-Level, geographic disparities emerged, with rural areas reporting higher zero-dose rates (16.59%) than urban settings (11.12%). Type of African region in sub-Sahra compounded these gaps: communities where children in central Africa had a zero-dose rate of 24.53%, compared to 7.09% for those living in southern Africa (Table 2).

**Table 2. Descriptive statistics of zero-dose vaccination among children by individual and community level characteristics (n = 58,443).**

| Variables | Zero-dose status | | Total [n (%)] | Chi-square test ($\chi^2$) | p-value |
|---|---|---|---|---|---|
| | Yes [n (%)] | No [n (%)] | | | |
| **Individual level characteristics** | | | | | |
| **Socio-demographic and economic characteristics** | | | | | |
| Gender | | | | 1.14 | 0.285 |
| Male | 4,328 (14.57) | 25,383 (85.43) | 29,711(50.84) | | |
| Female | 4,266 (14.85) | 24,466 (85.15) | 28,732 (49.16) | | |
| HH head gender | | | | 38.24 | <0.001 |
| Male | 7,044 (15.34) | 38,864(84.66) | 45,908 (78.55) | | |
| Female | 1,550 (12.37) | 10,985(87.63) | 12,535 (21.45) | | |
| Number of alive children | | | | 170.22 | <0.001 |
| One | 1639 (11.77) | 12284 (88.23) | 13,923 (23.82) | | |
| Two | 1703(13.39) | 11017 (86.61) | 12,720 (21.76) | | |
| Three | 1548 (14.99) | 8781 (85.01) | 10,329 (17.67) | | |
| Four or more | 3704 (17.25) | 17766 (82.75) | 21,470 (36.75) | | |
| Wealth index | | | | 1600.0 | <0.001 |
| Poorest | 3044 (22.91) | 10244 (77.09) | 13289 (22.74) | | |
| Poorer | 2244 (17.83) | 10344 (82.17) | 12588 (21.54) | | |
| Middle | 1570 (13.39) | 10157 (86.61) | 11727 (20.06) | | |
| Richer | 1032 (9.49) | 9840 (90.51) | 10872 (18.60) | | |
| Richest | 704 (7.07) | 9264 (92.93) | 9968 (17.06) | | |
| Media exposure | | | | 1500.00 | <0.001 |
| Exposed | 4571 (22.43) | 15806 (77.57) | 20377 (34.87) | | |
| Unexposed | 4023 (10.57) | 34043 (89.43) | 38066 (65.13) | | |
| **Obstetric related characteristics** | | | | | |
| Place of delivery | | | | 6800.00 | <0.001 |
| Home | 5276 (34.36) | 10080 (65.64) | 15,356 (26.27) | | |
| Health facility | 3318 (7.70) | 39769 (92.30) | 43,088 (73.73) | | |
| Delivery assistance | | | | 2100.00 | <0.001 |
| Skilled | 4,171 (9.42) | 40,109 (90.58) | 44,280 (75.02) | | |
| Unskilled | 3,109 (21.95) | 11,054 (78.05) | 14,163 (24.98) | | |
| ANC visits | | | | 6300.00 | <0.001 |
| No visits | 3503 (40.14) | 5225 (59.86) | 8,728 (14.93) | | |
| 1-3 visits | 2224 (13.99) | 13673 (86.01) | 15,897 (27.20) | | |
| 4 or more visits | 2867 (8.48) | 30951 (91.52) | 33,819 (57.87) | | |
| Postnatal checkup | | | | 907.06 | <0.001 |
| Yes | 1466 (8.18) | 16445 (91.82) | 17,911 (32.11) | | |
| No | 6,678 (17.63) | 31,190 (82.37) | 37,868 (67.89) | | |
| Number of TT vaccination | | | | 3700.00 | <0.001 |
| 0 | 4347 (28.45) | 10934 (71.55) | 15281 (26.15) | | |
| 1 | 1519 (9.73) | 14100 (90.27) | 15619 (26.73) | | |
| 2 or more | 2728 (9.90) | 24815 (90.10) | 27543 (47.13) | | |
| Birth order | | | | 226.73 | <0.001 |
| First | 1,594 (11.63) | 12,112 (88.37) | 13705 (23.45) | | |
| Second | 1,532 (13.00) | 10,259 (87.00) | 11791 (20.18) | | |
| Third | 1,378 (17.60) | 8,330 (82.40) | 9708 (16.61) | | |
| Fourth or more | 4,090 (11.63) | 19,149 (88.37) | 23238 (39.76) | | |

*(Continued)*

**Table 2.** (Continued)

| Variables | Zero-dose status | | Total | Chi-square test ($\chi^2$) | p-value |
|---|---|---|---|---|---|
| | Yes [n (%)] | No [n (%)] | [n (%)] | | |
| **Parent related characteristics** | | | | | |
| Maternal age | | | | 99.31 | <0.001 |
| 15-19 | 894 (19.14) | 3774 (80.86) | 4668 (7.99) | | |
| 20-34 | 5845 (13.98) | 35966 (86.02) | 41811 (71.54) | | |
| 35-49 | 1856 (15.51) | 10109 (84.49) | 11965 (20.47) | | |
| Maternal marital status | | | | 14.84 | <0.001 |
| In union | 42698 (85.02) | 7526 (14.98) | 50,223 (85.94) | | |
| Not in union | 7151 (87.00) | 1069 (13.00) | 8,220 (14.06) | | |
| Maternal educational status | | | | 300.00 | <0.001 |
| No education | 5081 (25.34) | 14970 (74.66) | 20,051 (34.31) | | |
| Primary | 2069 (10.73) | 17210 (89.27) | 19,279 (32.99) | | |
| Secondary or higher | 1445 (7.56) | 17668 (92.44) | 19,113 (32.70) | | |
| Maternal occupation | | | | 96.02 | <0.001 |
| Employed | 4715 (13.31) | 30712 (83.14) | 35,426 (60.62) | | |
| Unemployed | 3880 (16.86) | 19137 (86.69) | 23,017 (39.38) | | |
| Maternal autonomy | | | | 166.64 | <0.001 |
| Autonomous | 935 (10.28) | 8163 (89.72) | 9,098 (15.57) | | |
| Not autonomous | 7659 (15.52) | 41686 (84.48) | 49,345 (84.43) | | |
| Paternal education | | | | 1200.00 | <0.001 |
| No education | 5,396 (20.14) | 21,396 (79.86) | 26792 (45.84) | | |
| Primary | 1,636 (11.59) | 12,478 (88.41) | 14114 (24.15) | | |
| Secondary or higher | 1,563 (8.91) | 15,975 (91.09) | 17537 (30.01) | | |
| Paternal occupation | | | | 17.31 | <0.001 |
| Employed | 6508 (14.49) | 38405 (85.51) | 44914 (76.85) | | |
| Unemployed | 2086 (15.42) | 11,444 (84.58) | 13530 (23.15) | | |
| Distance to health facility | | | | 529.15 | <0.001 |
| Big problem | 4,289 (17.89) | 19,690 (82.11) | 23979 (41.03) | | |
| Not a big problem | 4,305 (12.49) | 30,159 (87.51) | 34464 (58.97) | | |
| **Community level characteristics** | | | | | |
| Residence | | | | 300.11 | <0.001 |
| Urban | 2,237 (11.12) | 17,884 (88.88) | 20,122 (34.43) | | |
| Rural | 6,357 (16.59) | 31,964 (83.41) | 38,322 (65.57) | | |
| Geographical region | | | | 1900.00 | <0.001 |
| Central | 1419 (24.53) | 4366 (75.47) | 5,784 (9.90) | | |
| Eastern | 2201 (8.75) | 22961 (91.25) | 25,162 (43.05) | | |
| Southern | 83 (7.09) | 1084 (92.91) | 1,167 (2.00) | | |
| Western | 4892 (18.58) | 21438 (81.42) | 26,330 (45.05) | | |

ANC: Antenatal Care, HH: Household, TT: Tetanus Toxoid.

## Prevalence of zero-dose vaccination among children aged 12–23 months in SSA

The prevalence of zero-dose vaccination among children aged 12–23 months in SSA was estimated at 13.3% with a 95% CI of 10.2–16.3%. The variation in the prevalence among the countries was high, with Guinea, Nigeria, Angola, and Côte

d'Ivoire having very high rates of 37.9% (CI: 35.3–40.4%), 35.2% (CI: 34.0–36.4%), 32.1% (CI: 30.4–34.0%), and 30.2% (CI: 28.1–32.3%), respectively. Contrary to the above, Rwanda recorded the lowest prevalence rate of 1.0% (CI: 0.8–1.2%), followed by Burundi (2.0%, CI: 1.2–2.3%), Gambia (2.0%, CI: 1.4–2.8%), and Zambia (2.1%, CI: 1.6–2.9%). High heterogeneity was seen in the data ($I^2 = 99.66\%$, $p < 0.001$), and hence subgroup analysis by important parameters was needed to be done to determine its probable source. The weights of the countries (3.51–3.60) reflect equal contributions from all countries (Fig 4).

We performed subgroup analysis between survey years and sub-regions. Accordingly, in the 2015–2019 surveys, the prevalence was 15.9% (CI: 10.4–21.3%; $I^2 = 99.75\%$, $p < 0.001$), and in the surveys of 2020–2024, it was 10.7% (CI: 7.4–14.1%; $I^2 = 99.44\%$, $p < 0.001$) with temporal difference, though not statistically significant as the estimate of one fell in the

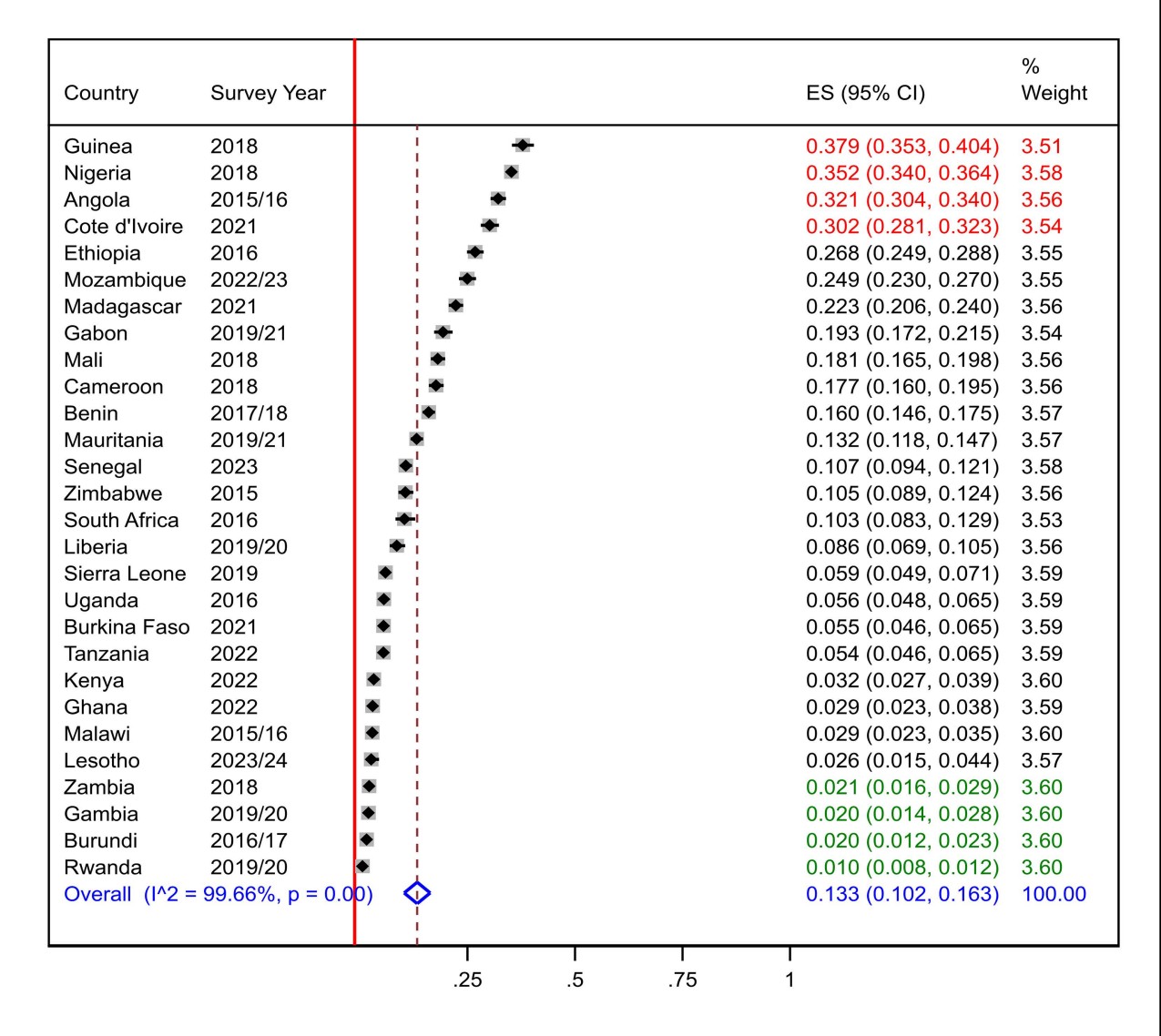

**Fig 4. Pooled prevalence of zero-dose vaccination among children aged 12-23 months in sub-Saharan Africa.**

95% CI of the other (Fig 5). Sub-regional analysis found Central SSA with the highest prevalence (23.0%, CI: 14.7–32.3%; I²= 98.60%), then Western SSA (15.5%, CI: 9.2–22.7%; I²= 99.70%), Eastern SSA (9.5%, CI: 7.5–12.5%; I²= 99.47%), and Southern SSA (4.7%, CI: 3.5–5.9%; I²= 99.50%) (Fig 6). The results affirm that prevalence varies in terms of geographic setting—rather than survey year.

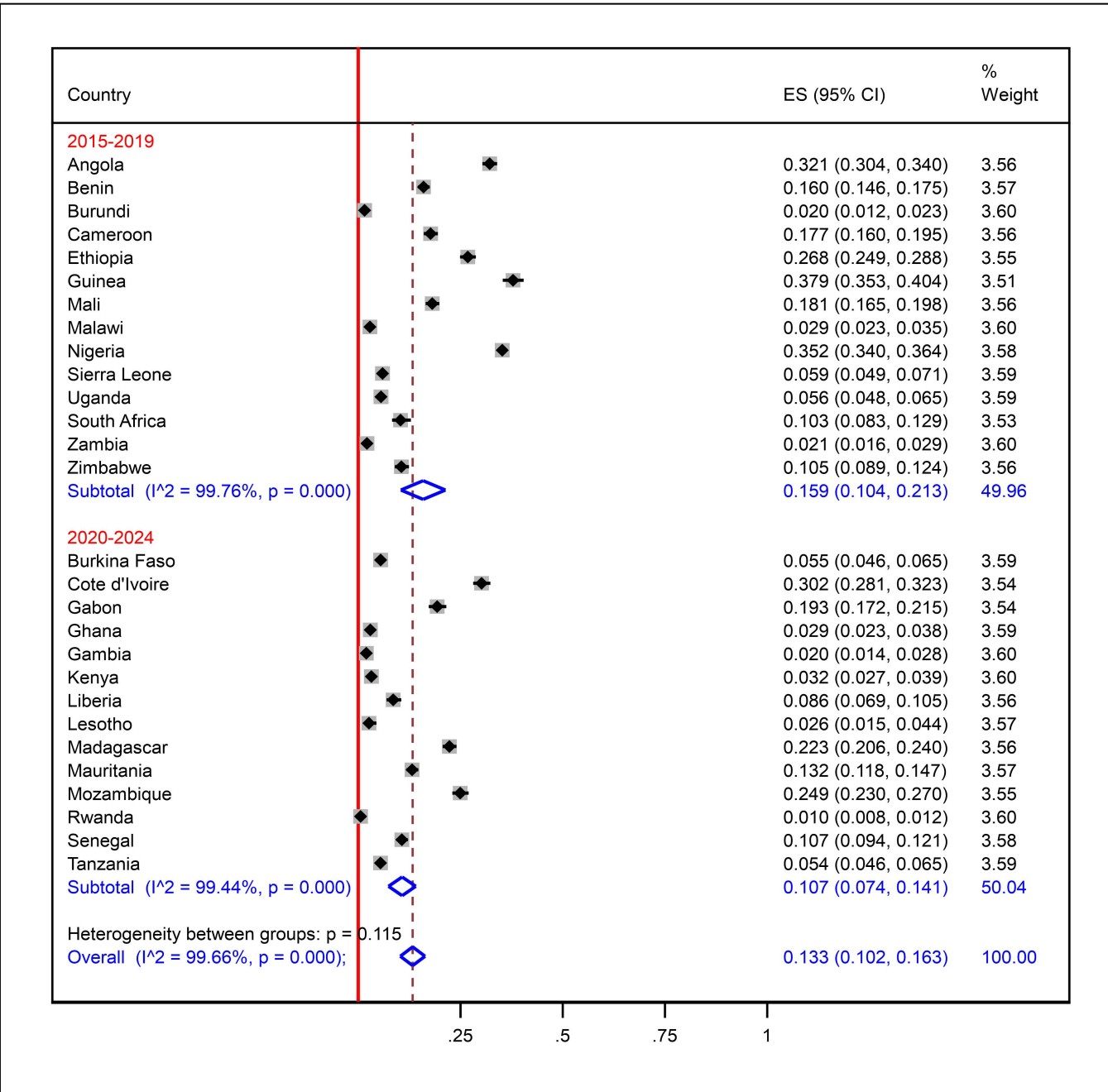

**Fig 5. Subgroup analysis of prevalence of zero-dose vaccination among children aged 12-23 months in sub-Saharan Africa by survey year.**

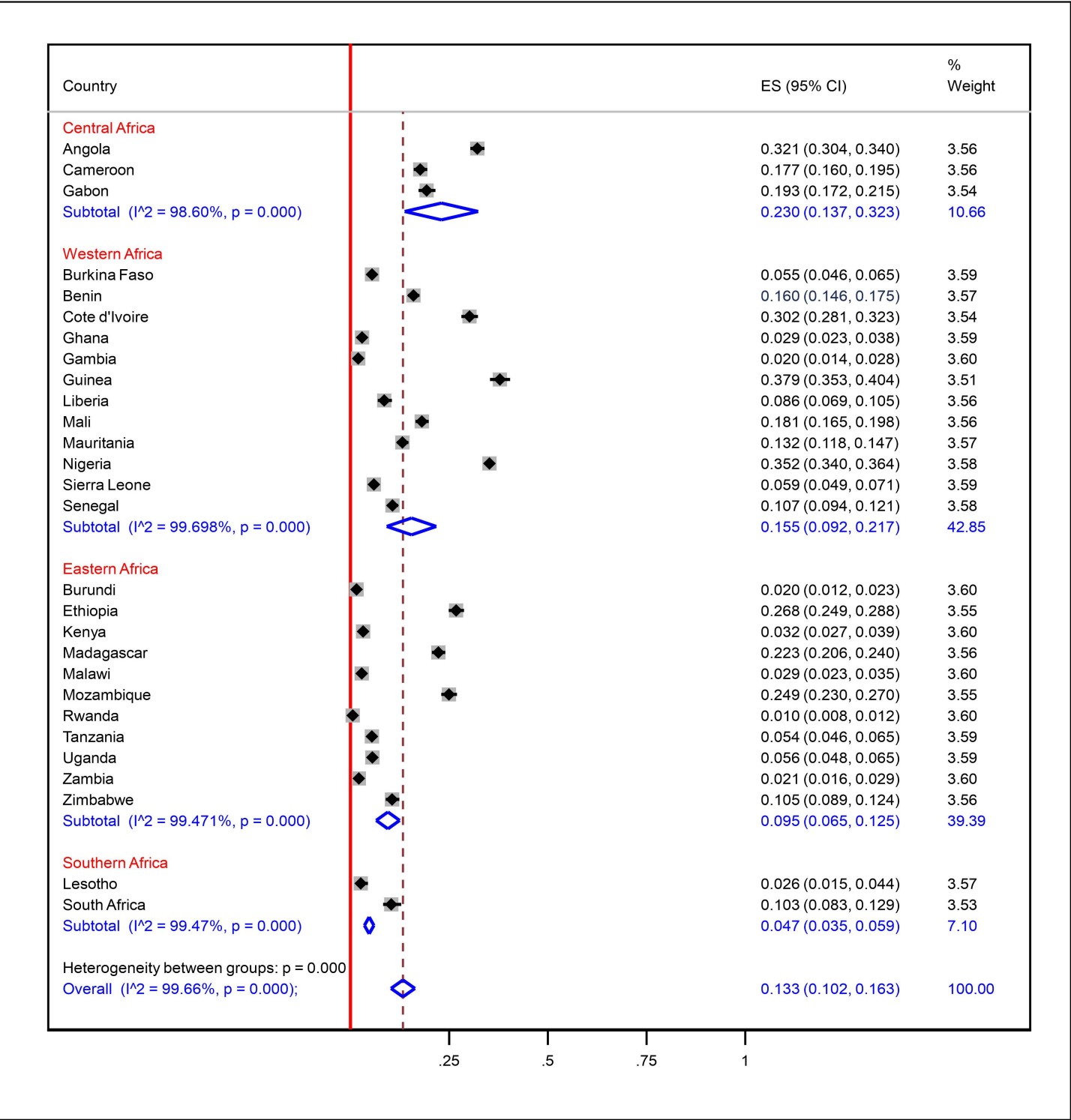

**Fig 6. Subgroup analysis of prevalence of zero-dose vaccination among children aged 12-23 months in sub-Saharan Africa by geographical regions.**

**Spatial autocorrelation of zero-dose vaccination among children aged 12–23 months in SSA**

Spatial autocorrelation test of zero-dose vaccination for children in SSA indicates that there were significant spatial patterns. The test result was a Moran's Index of 0.109725, reflecting positive correlation of the zero-dose rate distribution across the region. The Z-score of 6.397457 further supports this result, as it is far beyond the cutoff for randomness, and its corresponding p-value of less than 0.001 indicates that the observed clustering of zero-dose vaccination among children in SSA was highly unlikely to have occurred by chance. Such significant statistical evidence demonstrates that zero-dose vaccination regions are not randomly scattered or distributed but rather are more clustered (Fig 7). The finding favors the significance of further spatial analysis of zero-dose vaccination in SSA.

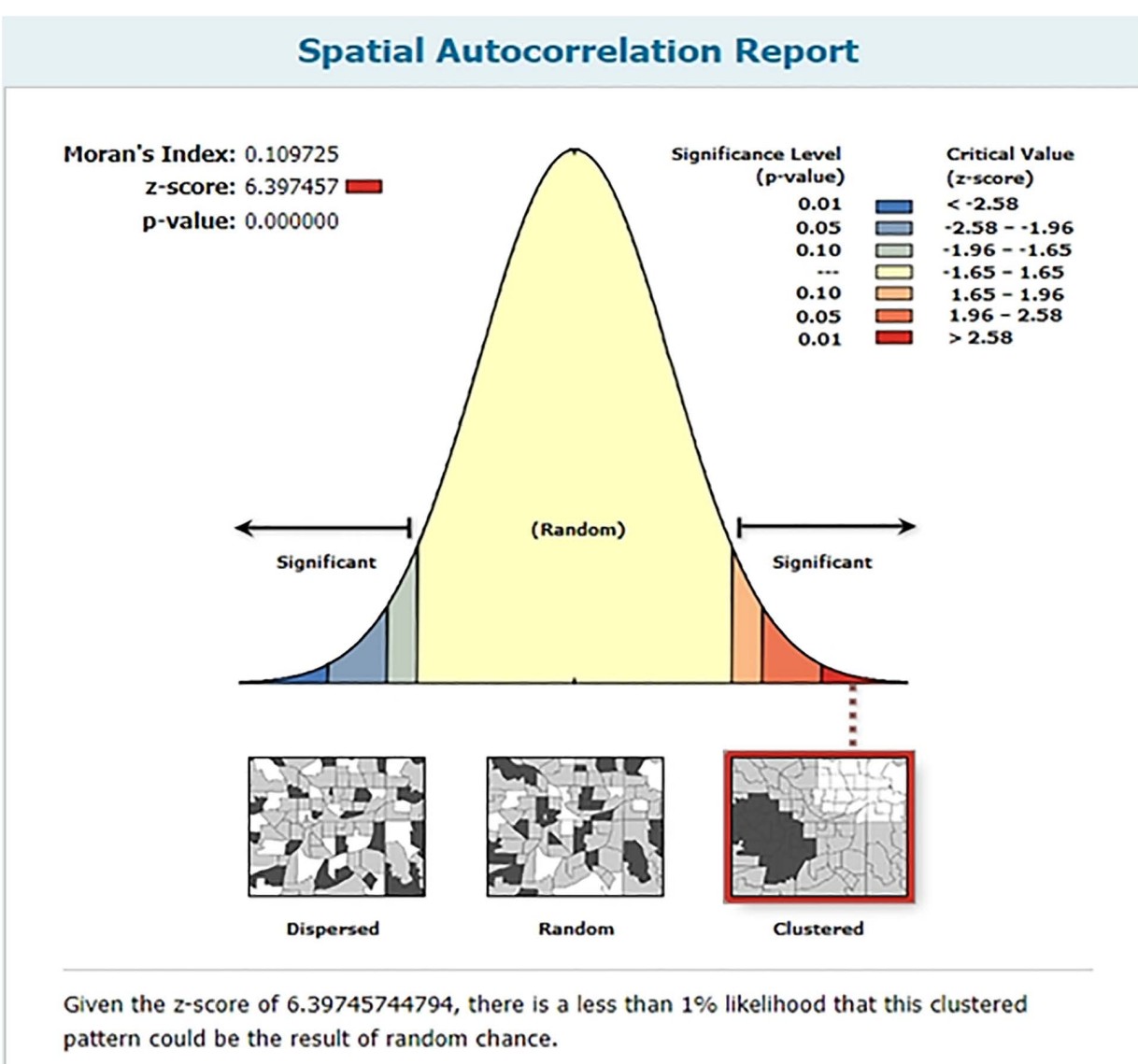

**Fig 7. Spatial distribution of zero-dose vaccination among children aged 12-23 months in SSA.**

## Optimized hotspot analysis of zero-dose vaccination among children aged 12–23 months in SSA

An optimized hot spot analysis was used to detect the hot and cold spots for zero-dose vaccination among children in SSA. Following the optimized hot spot analysis, there were 8,702 output features (spots) statistically significant based on false discovery rate (FDR) correction for multiple testing and spatial dependence. About 6.6% of features had less than 8 neighbors based on distance band of 69,637.00 meters. Accordingly, red output features represent hot spots where high zero-dose vaccination values cluster and green output features represent cold spots where low zero-dose vaccination among children in SSA values cluster. The hot spot areas of zero-dose vaccination were observed at the clusters in most parts of Nigeria, Angola, Ethiopia, Madagascar, Guinea, Cote d'Ivoire, southern border of Mauritania, and southeastern parts of Mali and Mozambique. In contrast, clusters at Burkina Faso, Burundi, Ghana, Kenya, Lesotho, Malawi, Senegal, Uganda, central part of Ethiopia and East of Zambia were cold spot areas of zero-dose vaccination among children in SSA (Fig 8).

## Spatial scan statistical analysis of zero-dose vaccination among children aged 12–23 months in SSA

The SaTScan spatial cluster analysis, which was conducted using 40% of the total population as the maximum size of the cluster window, identified 15 statistically significant clusters for zero-dose vaccination. A most likely (primary) cluster and 14 secondary clusters. The most likely (primary) cluster window was located at 13.239833°N latitude and 7.9633830E longitude, with a radius of 647.58 km. Within this cluster window, there were 685 enumeration areas, a population of 4,080, and 1,949 cases of zero-dose vaccination. The RR within this primary cluster was 3.91, indicating that the risk of zero-dose vaccination was 3.91 times higher for individuals living within this 647.58 km radius compared to those outside of it. The p-value for this primary cluster was highly significant at less than 0.001, suggesting that the elevated risk was unlikely to have occurred by chance (Table 3). While the most likely cluster window was observed to cover the most parts of Nigeria and extended to the Niger; the secondary SaTScan clusters encircled areas in the most parts of Angola, Ethiopia, Madagascar, Nigeria, Guinea, Cote d'Ivoire, southern border of Mauritania, and southeastern parts of Mali and Mozambique (Fig 9).

## Interpolation of zero-dose vaccination among children aged 12–23 months in SSA

The output of Kriging interpolation analysis illustrated in Fig 10 shows predicted locations for high and low zero-dose vaccination values throughout SSA. The color gradient on the map ranges from low values of prediction shaded in green to high values depicted in orange to red. The high values were primarily concentrated in specific geographical locations, including the majority of Angola, Niger, northern Nigeria, eastern regions of Ethiopia, east of Mali, Somalia, Northwest of Guinea, Southwest of Congo and Northern half of South Sudan. The predicted value of zero-dose vaccination ranges from a low of 0 to a high of 1 (Fig 10).

## Determinants of zero-dose vaccination among children aged 12–23 months in SSA

**Random effect.** The multilevel spatial analysis revealed substantial variation between communities. In the null model (Model 0), the between-cluster variance was 1.27, which decreased to 0.81 in the full model (Model III), indicating that some of the heterogeneity was explained by the included covariates. The ICC in Model 0 was 27.85%, suggesting that more than one-quarter of the total variation in zero-dose vaccination status was attributable to differences between clusters—strongly justifying the use of a multilevel modeling approach. The modest decline in ICC across successive models suggests that the covariates explain part of this between-community variation.

In the null model, the MOR was 2.92, quantifying the unexplained cluster-level heterogeneity. This implies that, when comparing two children with the same characteristics from different clusters, the child from the higher-risk cluster would have nearly three times the odds of being zero-dose solely due to unmeasured contextual factors.

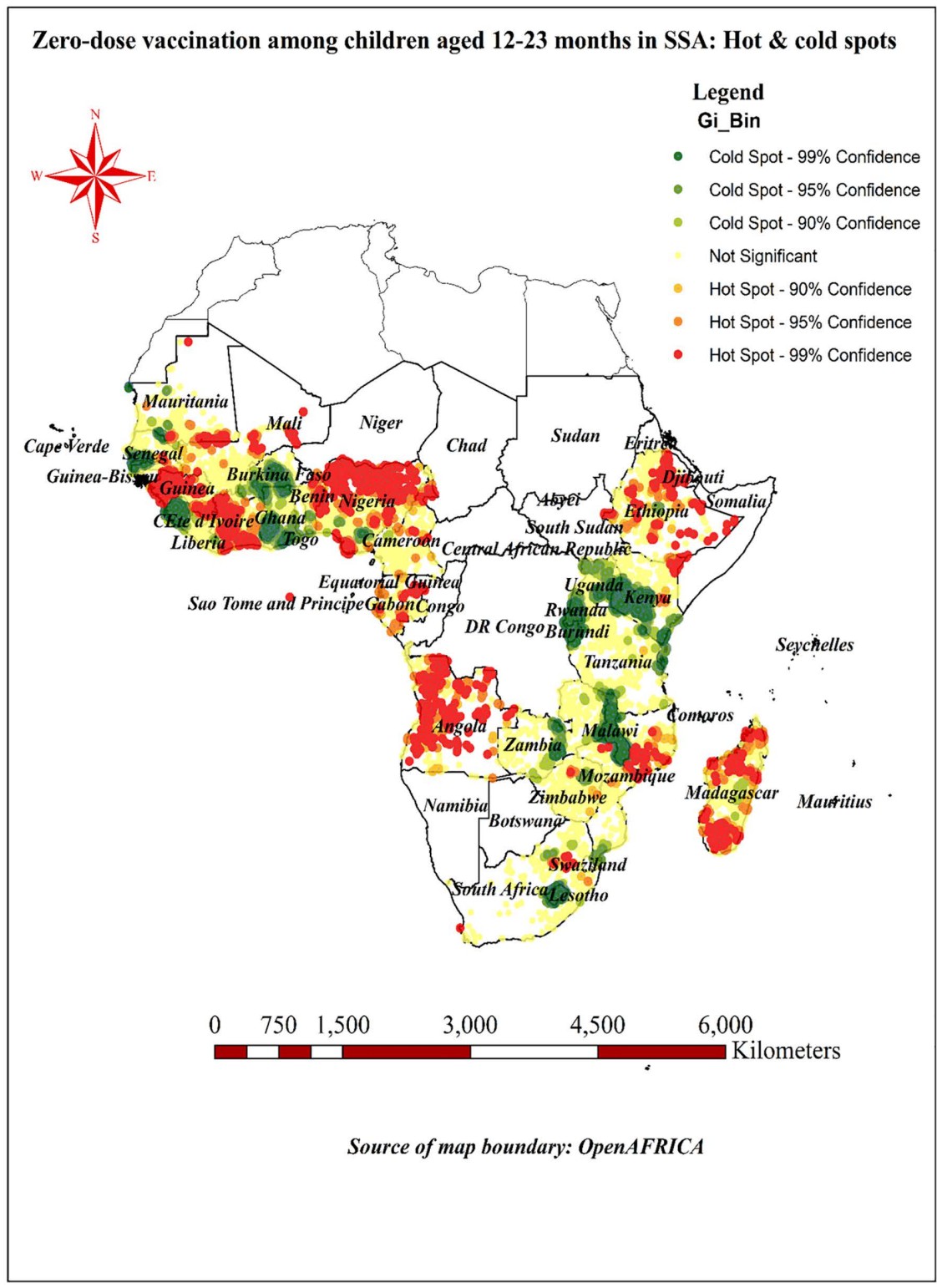

**Fig 8. Spatial distribution of zero-dose vaccination among children aged 12-23 months in SSA; Map created using Africa shapefiles – Admin Level 0 under Open Data Commons Open Database License from openAFRICA.**

**Table 3. Spatial scan statistics analysis of zero-dose vaccination among children aged 12-23 months in SSA.**

| Cluster | N | Latitude | Longitude | Radius (KM) | Population | Cases | RR | LLR | P value |
|---|---|---|---|---|---|---|---|---|---|
| Primary cluster | 685 | 13.239833N | 7.963383E | 647.58 | 4080 | 1949 | 3.91 | 1390.93 | <0.001 |
| Secondary cluster 1 | 436 | 13.171088N | 8.224928E | 476.34 | 2898 | 1556 | 4.24 | 1290.41 | <0.001 |
| Secondary cluster 2 | 370 | 11.867772S | 18.007725E | 584.41 | 1453 | 614 | 3.02 | 332.04 | <0.001 |
| Secondary cluster 3 | 179 | 11.919215N | 12.255909W | 213.54 | 554 | 290 | 3.65 | 217.46 | <0.001 |
| Secondary cluster 4 | 344 | 6.201689N | 5.275584W | 250.05 | 1350 | 459 | 2.39 | 160.90 | <0.001 |
| Secondary cluster 5 | 824 | 19.564538S | 44.457553E | 1001.08 | 3351 | 861 | 1.83 | 146.87 | <0.001 |
| Secondary cluster 6 | 74 | 17.269259S | 37.822882E | 256.66 | 463 | 221 | 3.30 | 143.25 | <0.001 |
| Secondary cluster 7 | 349 | 8.888553N | 40.744565E | 467.11 | 1374 | 440 | 2.24 | 134.37 | <0.001 |
| Secondary cluster 8 | 55 | 23.203265 S | 43.629900E | 250.40 | 319 | 165 | 3.57 | 120.79 | <0.001 |
| Secondary cluster 9 | 64 | 9.332164N | 3.710396 E | 142.59 | 372 | 155 | 2.87 | 79.64 | <0.001 |
| Secondary cluster 10 | 174 | 12.304137S | 49.279051E | 628.41 | 596 | 177 | 2.04 | 43.93 | <0.001 |
| Secondary cluster 11 | 183 | 16.396384N | 6.515007W | 339.33 | 710 | 196 | 1.90 | 39.705 | <0.001 |
| Secondary cluster 12 | 57 | 9.235405 N | 8.335456W | 149.02 | 155 | 69 | 3.04 | 39.576 | <0.001 |
| Secondary cluster 13 | 3 | 18.076831N | 15.997464W | 0.16 | 15 | 15 | 6.81 | 28.76 | <0.001 |
| Secondary cluster 14 | 701 | 4.432267N | 14.370071E | 703.64 | 2875 | 546 | 1.31 | 20.63 | 0.001 |

N: Number of enumeration areas in a cluster, RR: Relative Risk, LLR: Log Likelihood Ratio.

The PCV showed that 36.22% of the between-cluster variation was explained by individual- and community-level covariates in the full model. This demonstrates the contribution of these determinants to explaining zero-dose vulnerability.

From a spatial perspective, the model indicated notable reductions in spatial heterogeneity. The Matérn range, which defines the distance over which spatial correlation persists, was 3.52 degrees (~391 km) in the null model. This means that clusters located within 391 km of each other tend to exhibit more similar zero-dose patterns due to shared spatial factors. In Model III, the range decreased to 2.92 degrees (~324 km), suggesting that once covariates were included, spatial clustering occurred over shorter distances. A spatial variance, a measure portion of total variation in the zero-dose that was attributable to spatially structured differences between geographic areas, was 2.37 in model 0. The value 2.37 indicates that geographic location alone explains a meaningful portion of the variation in zero-dose vaccination rates and it implied that where a child lives significantly influences their likelihood of being zero-dose. In the full model it decreased to 1.02, indicating that much of the previously unexplained spatial variation was accounted for by the covariates included in the full model.

As the models progressed from Model 0 to Model III, overall fit improved significantly. This was evidenced by a reduction in the maximum likelihood criterion (from 18,322.97 to 16746.40), as well as a ~3,000-point drop in the AIC and BIC. These improvements underscore that Model III provides a more parsimonious framework for understanding the determinants of zero-dose vaccination (Table 4).

**Fixed effect.** Following chi-squared association testing, bivariate analysis, and multicollinearity assessment, the variables gender of the child, type of delivery assistance, and birth order were excluded from the multivariable regression. The variable gender of the child showed no significant association with the outcome in the chi-squared test and had a p-value greater than 0.25 in the bivariate analysis. Type of delivery assistance and birth order exhibited multicollinearity with place of delivery and number of children in the household, respectively. After adjusting for relevant covariates, our analysis identified several factors significantly associated with zero-dose vaccination among children. These included the household wealth index, place of delivery, media exposure, antenatal care visits, tetanus toxoid vaccination, maternal education, perceived distance to health facilities and place of residence were significantly associated with zero-dose vaccination among children in SSA.

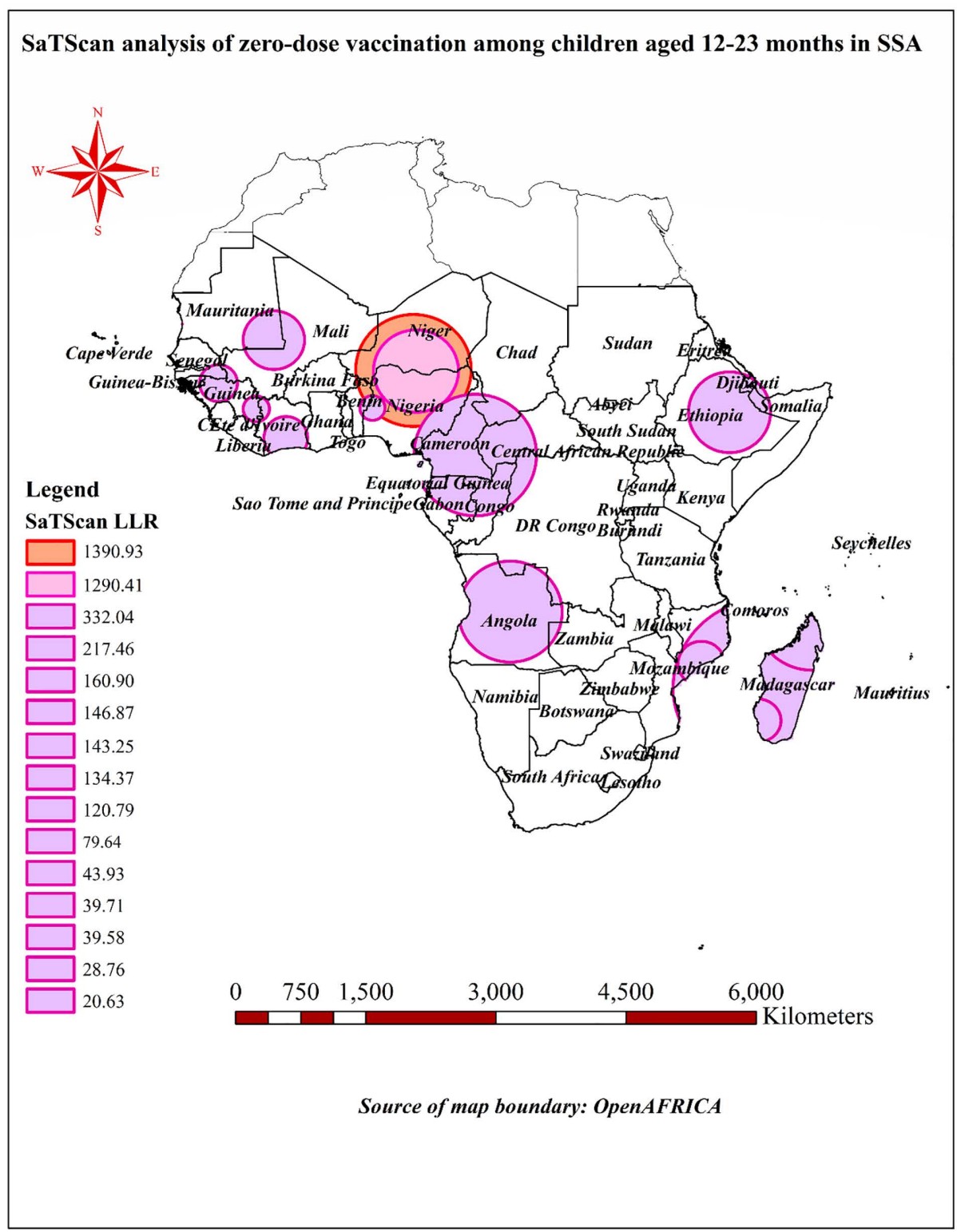

**Fig 9. Spatial scan statistical analysis of zero-dose vaccination among children aged 12-23 months in SSA; Map created using Africa shape-files – Admin Level 0 under Open Data Commons Open Database License from openAFRICA.**

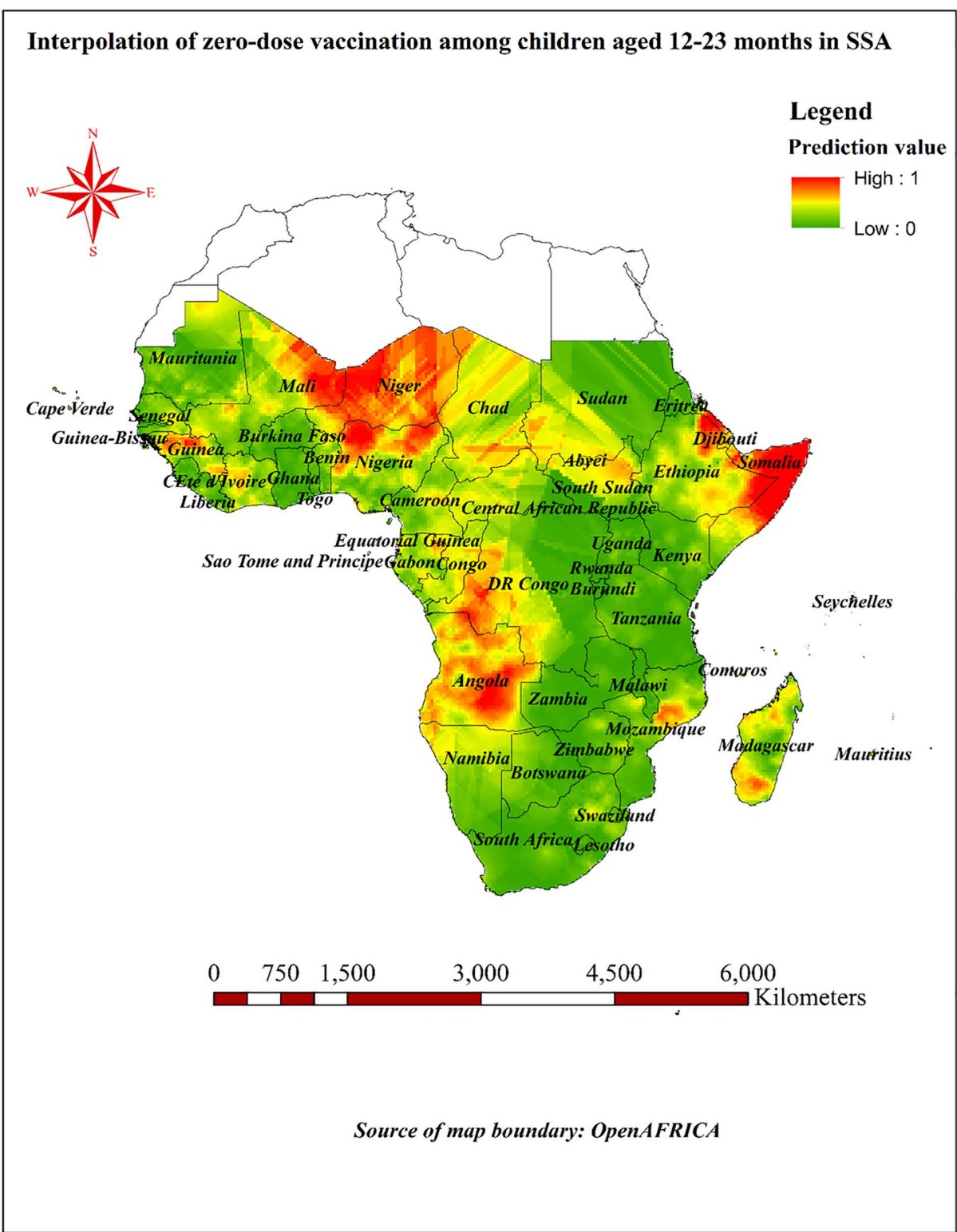

**Fig 10. Prediction of zero-dose vaccination among children aged 12-23 months throughout SSA; Map created using Africa shapefiles – Admin Level 0 under Open Data Commons Open Database License from openAFRICA.**

Notably, children from the poorest households had 30% higher odds of being zero-dose vaccinated compared to those from the richest households (AOR = 1.30; 95% CI: 1.10–1.53). Media exposure also played a role. Children from households with no media exposure had 25% higher odds of being zero-dose (AOR = 1.25; 95% CI: 1.14–1.35) compared to children from households with media exposure. Regarding place of delivery, children born at home had 2. 60 times higher odds of being zero-dose vaccinated than those born in health facilities (AOR = 2.60; 95% CI: 2.39–2.83).

Maternal healthcare utilization (ANC and TT vaccination) was significantly associated with zero-dose vaccination. Children whose mothers had no ANC visits had 2.41 times higher odds of being zero-dose (AOR = 2.41; 95% CI: 2.17–2.68), and those with only 1–3 visits had 37% higher odds (AOR = 1.37; 95% CI: 1.26–1.49), compared to those whose mothers had four or more visits. Similarly, children whose mothers received no TT vaccination during pregnancy had 63% higher odds (AOR = 1.63; 95% CI: 1.49–1.79). Maternal education was also associated with zero-dose vaccination. Specifically, maternal education was negatively associated with zero-dose vaccination. In this respect, children of mothers with no formal education within the same cluster had 1.81% higher odds of being zero-dose (AOR = 1.81; 95% CI: 1.62–2.03), and those with only primary education had 28% higher odds (AOR = 1.28; 95% CI: 1.16–1.43), compared to children of mothers with secondary or higher education. Moreover, the perceived distance to a health facility was significantly associated with zero-dose. Accordingly, Children from households where distance was considered a "big problem" had 13% higher odds of zero-dose vaccination (AOR = 1.13; 95% CI: 1.05–1.21). Finally, the odds of zero dose vaccination were 1.25 times higher among rural resident children compared to the urban resident ones (AOR = 1.25; 95% CI: 1.07–1.67) (Table 4).

## Discussion

This study revealed the prevalence, spatial variation, and individual- & community-level factors of zero-dose vaccination among children aged 12–23 months in sub-Saharan Africa using data from the 2015–2024 DHS with the application of a multilevel spatial model which accounts for the hierarchical nature of the data with its spatial dimension.

In this study, the overall prevalence of zero-dose vaccination among children aged 12–23 months in sub-Saharan Africa was estimated to be 13% with a 95% CI (10%, 16%). Interestingly, our prevalence is in agreement with the previously reported in a systematic review of 204 countries that had essentially the same range with zero-dose prevalence ranging from 13.4% to 15.9% [66]. Our prevalence also aligns closely with data from specific population groups. Among children living in poor urban communities, the zero-dose rate was 12.6%, while in rural areas it was slightly higher at 14.7% according to a study across 97 low- and middle-income countries [42]. The prevalence also aligned with what has been observed in individual countries as well: In India, a study reported a zero-dose prevalence of 10.1% [10] and a study in Pakistan reported 10.6% prevalence of zero-dose vaccination [67]. The consistency across different regions and socio-economic settings shows a persistent challenge in reaching all children with life-saving vaccines.

However, the prevalence observed in our study was notably higher than what has been reported in some previous research conducted in low- and middle-income countries. A study that analyzed data from 92 low-and middle income countries estimated the pooled prevalence of zero-dose vaccination among children aged 12–23 months was 7.7%, far lower than our study. Similarly, a cross-sectional study covering 82 low- and middle-income countries found that 7.5% of children in the same age group were zero-dose [68]. In addition, a study examining the path to full immunization in 43 low- and middle-income countries reported that 9% of children between 12 and 23 months were zero-dose [69]. Again, this figure is lower than the prevalence reported in our study. The discrepancy could be most likely due to the reason that our study determined zero-dose when a baby was not vaccinated for DPT1 as per the current WHO definition, while the earlier ones determined it when a baby was never vaccinated for any dose of essential childhood vaccines [68,69]. Moreover, the nature of the data used for estimating the prevalence of zero-dose might have also played a role in the discrepancy in the prevalences: these previous studies used both DHS and Multiple Indicator Cluster Surveys (MICS) data, while this study was solely based on DHS data due to the lack of spatial information, which was needed by our spatial analysis as a key aspect of the study, in the majority of MICS data. Although DHS and MICS are highly comparable surveys and

**Table 4. A multilevel spatial analysis of zero-dose vaccination among children aged 12-23 months.**

| Fixed Effect with AOR at 95% CI | | | |
|---|---|---|---|
| **Parameters** | **Model I** | **Model II** | **Model III** |
| HH head gender | | | |
| Male | 0.96 (0.88, 1.06) | | 0.97 (0.88, 1.06) |
| Female | 1.00 | | 1.00 |
| Number of alive children | | | |
| One | 1.00 | | 1.00 |
| Two | 0.97 (0.87, 1.07) | | 0.95 (0.86, 1.07) |
| Three | 1.13 (1.01, 1.27) | | 1.11 (0.97, 1.24) |
| Four or more | 0.97 (0.88, 1.08) | | 0.96 (0.87, 1.08) |
| Wealth index | | | |
| Poorest | 1.34 (1.15, 1.56) | | **1.30 (1.10, 1.53) *** |
| Poorer | 1.17 (1.01, 1.35) | | 1.13 (0.97, 1.32) |
| Middle | 1.12 (0.98, 1.29) | | 1.10 (0.95, 1.27) |
| Richer | 0.97 (0.85, 1.11) | | 0.96 (0.84, 1.10) |
| Richest | 1.00 | | 1.00 |
| Media exposure | | | |
| Exposed | 1.00 | | 1.00 |
| Unexposed | 1.24 (1.15, 1.34) | | **1.25 (1.14, 1.35) *** |
| Place of delivery | | | |
| Home | 2.59 (2.39, 2.82) | | **2.60 (2.39, 2.83) *** |
| Health facility | 1.00 | | 1.00 |
| ANC visits | | | |
| No visits | 2.51 (2.26, 2.79) | | **2.41 (2.17, 2.68) *** |
| 1-3 visits | 1.38 (1.27, 1.49) | | **1.37 (1.26, 1.49) *** |
| 4 or more visits | 1.00 | | 1.00 |
| Postnatal checkup | | | |
| Yes | 1.00 | | 1.00 |
| No | 1.06 (0.95, 1.11) | | 1.05 (0.93, 1.10) |
| Number of TT vaccination | | | |
| 0 | 1.66 (1.51, 1.82) | | **1.63 (1.49, 1.79) *** |
| 1 | 1.05 (0.96, 1.14) | | 1.04 (0.96, 1.14) |
| 2 or more | 1.00 | | 1.00 |
| Maternal age | | | |
| 15-19 | 1.07 (1.03, 1.10) | | 0.97 (0.93, 1.02) |
| 20-34 | 0.99 (0.91, 1.08) | | 0.99 (0.90, 1.08) |
| 35-49 | 1.00 | | 1.00 |
| Maternal marital status | | | |
| In union | 1.00 | | 1.00 |
| Not in union | 0.89 (0.76, 1.03) | | 0.89 (0.76, 1.04) |
| Maternal educational status | | | |
| No education | 1.82(1.62, 2.04) | | **1.81 (1.62, 2.03) *** |
| Primary | 1.28 (1.16, 1.42) | | **1.28 (1.16, 1.43) *** |
| Secondary or higher | 1.00 | | 1.00 |

*(Continued)*

**Table 4.** (Continued)

| Fixed Effect with AOR at 95% CI | | | |
|---|---|---|---|
| **Parameters** | **Model I** | **Model II** | **Model III** |
| Maternal occupation | | | |
| Employed | 1.00 | | 1.00 |
| Unemployed | 1.05 (1.02, 1.11) | | 1.02 (0.98, 1.09) |
| Maternal autonomy | | | |
| Autonomous | 1.00 | | 1.00 |
| Not autonomous | 1.06 (0.96, 1.18) | | 1.06 (0.95, 1.17) |
| Paternal education | | | |
| No education | 1.16 (0.95, 1.26) | | 1.14 (0.93, 1.24) |
| Primary | 1.11 (0.91, 1.23) | | 1.10 (0.90, 1.22) |
| Secondary or higher | 1.00 | | 1.00 |
| Paternal occupation | | | |
| Employed | 1.00 | | 1.00 |
| Unemployed | 1.06 (1.02, 1.08) | | 0.98 (0.93, 1.05) |
| Distance to HF | | | |
| Big problem | 1.14 (1.05, 1.22) | | **1.13 (1.05, 1.21) *** |
| Not a big problem | 1.00 | | 1.00 |
| Residence | | | |
| Urban | | 1.00 | 1.00 |
| Rural | | 1.99 (1.79, 2.21) | **1.25 (1.07, 1.67) *** |
| Geographical region | | | |
| Central | | 6.80 (2.36, 19.6) | 1.97 (0.86, 4.53) |
| Eastern | | 0.98 (0.37, 2.57) | 0.47 (0.21, 1.05) |
| Southern | | 1.00 | 1.00 |
| Western | | 3. 59 (1.30, 9.94) | 1.21 (0.53, 2.73) |
| **Radom Effect** | | | |
| **Metrics** | **Model 0** | **Model I** | **Model II** | **Model III** |
| Variance | 1.27 | 1.04 | 1.21 | 0.81 |
| ICC (%) | 27.85 | 24.01 | 26.89 | 19.76 |
| MOR | 2.92 | 2.63 | 2.84 | 2.35 |
| PCV (%) | Reference | 18.11 | 4.72 | 36.22 |
| Matern range (°) | 3.52 | 3.44 | 3.11 | 2.93 |
| Spatial variance | 2.37 | 1.23 | 1.88 | 1.02 |
| ML criterion | 18322.97 | 16760.66 | 18225.59 | 16746.40 |
| AIC | 36653.93 | 33581.33 | 36467.18 | 33560.79 |
| BIC | 36689.87 | 33866.24 | 36539.05 | 33850.84 |

**Key:** AOR: Adjusted Odds Ratio, AIC: Akaike's Information Criterion, ANC: Antenatal care, BIC: Bayesian Information Criterion, CI: Confidence Interval, HF: Health Facility, HH: Household, ICC: Intra-class Correlation Coefficient, ML = Maximum likelihood at convergence, MOR: Median Odds Ratio, OR= Odds Ratio, PCV: Proportional Change in Variation, TT: Tetanus Toxoid, * p < 0.05, mean VIF = 1.36 with min = 1.04 and max = 1.84.

use almost identical immunization modules, there is a key difference: MICS includes orphans and foster children in the vaccination module, while DHS does not [70]. DHS collects vaccination information only from biological mothers, while MICS also includes other primary caregivers [70,71]. That said, these comparisons highlight a concerning gap in

immunization coverage in our study setting, suggesting that children in this region may face greater barriers to accessing routine vaccination services than those in many other low- and middle-income countries.

The marked variation in prevalence of zero-dose vaccination across sub-Saharan Africa, high rates in Guinea (37.9%), Nigeria (35.2%), Angola (32.1%), and Côte d'Ivoire (30.2%), yet encouragingly low rates in Rwanda (1.0%), Burundi (2.0%), Gambia (2.0%), and Zambia (2.1%)—reflect a complex tapestry of contextual factors. While shared challenges like geographic inaccessibility amplify vulnerability, the distinct historical, socioeconomic, and health system trajectories of each nation may critically shape outcomes. Countries achieving remarkable success, such as Rwanda, owe their progress to sustained investments in decentralized primary healthcare. Rwanda's network of community health workers, bolstered by performance-based financing, has driven near-universal immunization coverage by bridging last-mile gaps [72,73]. Similarly, Zambia leveraged culturally resonant community engagement and health education to reduce zero-dose numbers, particularly in remote areas [74]. Burundi and Gambia have demonstrated how political commitment to primary healthcare expansion can yield tangible gains [75,76]. Conversely, Nigeria's struggles stem from severe health system fragmentation, vaccine shortages in northern states, and insurgent conflicts displacing populations [77,78]. In Angola, decades of civil war eroded infrastructure and cold-chain capacity, perpetuating coverage gaps despite peace [79,80]. Guinea's health system, weakened by political instability and the 2014–2016 Ebola epidemic, faces acute workforce and infrastructure shortages [81]. In this respect, Guinea has low key health personnel for vaccination to population ratio of 3.71 Nursing and Midwifery personnel to 10,000 population as compared to other countries for example 21.5 of Zambia [82]. Côte d'Ivoire has made progress but still contends with subnational disparities and logistical hurdles in reaching remote populations [83].

The spatial autocorrelation of this study disclosed that the distribution of zero-dose vaccination was spatially clustered throughout This finding is in line with previous studies on zero-dose vaccination in low- and middle-income countries SSA [10,38–40,67,84–86,87]. In this regard, a study that compiled geolocated data on vaccination coverage and associated risk factors in six low and middle-income countries revealed substantial variation in the spatial distribution of zero-dose vaccination among children, highlighting the most vulnerable areas [40]. According to the optimized hot spot analysis of this study—supported by SaTScan analysis and kriging interpolation results—the most parts of Nigeria, Angola, Ethiopia, Madagascar, Guinea, Cote d'Ivoire, southern border of Mauritania, and southeastern parts of Mali and Mozambique were found to be hot spot areas of zero-dose vaccination among children in SSA. We also found high prevalence of zero-dose vaccination among children in these countries, which could be additional evidence of regional obstacles to receiving child-hood vaccination services. It has been evidenced that the spatial clustering and regional disparities in zero-dose children are attributed to a complex interplay of systemic, geographic, and socio-political factors. A significant proportion of these children reside in fragile, conflict-affected, or remote areas where health systems are either weak or disrupted, limiting access to routine immunization services [88,89]. In addition, geographic isolation, poor infrastructure, and long distances to health facilities remain persistent challenges for vaccination, particularly in rural and underserved urban settlements [89]. Moreover, hot spots of zero-dose children were reported to be concentrated in communities where socioeconomic inequities such as poverty and illiteracy rates are common [90]. These challenges are compounded by funding shortfalls, misinformation, and humanitarian crises, all of which strain immunization systems and contribute to the observed spatial variation in vaccine coverage [90]. Therefore, given the broad reach of immunization programs, enhancing equity in immunization coverage through enhanced service delivery and outreach may improve the delivery of additional health interventions and preventative measures aimed at children [91].

The multilevel spatial analysis of our study revealed that the household wealth index, place of delivery, media exposure, antenatal care visits, tetanus toxoid vaccination, maternal education, perceived distance to health facilities and place of residence were significantly associated with zero-dose vaccination among children in SSA.

In agreement with previous studies [34,3868,92], household wealth quintile was significantly associated with zero-dose vaccination. The children from the poorest households had higher likelihood of zero-dose vaccination compared to

those from the richest households. This could be due to the reason that families with greater financial resources often have better access to quality healthcare services than families with lower financial resources [93]. This advantage means that children from wealthier households are more likely to receive vaccinations early in life compared to those from poorer families. Economic and social standing can play a powerful role in shaping health outcomes, especially when it comes to preventive care like immunization [94]. A study has also shown that disparities in household wealth can significantly impact whether children get vaccinated on time [93]. These inequalities reflect real barriers faced by caregivers in less affluent communities, who may struggle with transportation, time off work, or even awareness about vaccination schedules.

In this study, lack of media exposure was positively associated with zero-dose vaccination among children. Accordingly, children from households with no media exposure had higher odds of zero-dose vaccination compared to children from households with media exposure. This find aligns with existing evidence about media exposure and childhood vaccination [95,96]. This could be due to the reason that mass media interventions have the potential to create awareness about role of vaccination in prevention of disease and increase immunization coverage [97]. It plays a crucial role in disseminating information about childhood vaccination, thereby facilitating behavioral change toward vaccination practices [96]. Additionally, a lack of sufficient information about vaccines is associated with an increased likelihood of delaying or refusing vaccinations [39]; those who feel uninformed are particularly susceptible to vaccine hesitancy [98].

Antenatal care visit was associated with zero-dose vaccination. Specifically, the odds of zero-dose vaccination were higher among children whose mothers had no ANC visits and had 1–3 visits, compared to those whose mothers had four or more visits. This finding is in line with findings from several previous studies [68,99–105]. This could be as a result of the counselling and instruction mothers received about the value of postnatal visits and activities during ANC visits [104]. The ANC visits offer women an opportunity to learn about immunizations, and mother education plays a significant role in ensuring that children receive the necessary vaccinations [105]. Hence, public health services should continue to prioritize ANC visits and enhance maternal awareness in order to raise childhood immunization rates and address zero-dose children.

In addition, the odds of zero-dose vaccination were higher among children born at home compared to those born at health facility. The similar findings were reported from earlier studies [106–109]. This could be explained by the reason that in addition to the potential of lower maternal mortality and early newborn deaths associated with delivery-related events, health facility delivery encourages new mothers to vaccinate their newborns after delivery [107]. It is plausible that mothers who deliver at the health facility interact with the healthcare system in a way that encourages their children to seek care in the coming years and in contrast mothers give births at home fail to miss the opportunity.

In this study, receipt of TT vaccination was inversely associated with zero-dose vaccination. The children whose mothers received no TT vaccination during pregnancy had higher odds of zero-dose vaccination compared to children whose mothers received two or more tetanus toxoid containing vaccines. Previous studies have also witnessed similar finding [68,110]. This could be due to the fact that both are influenced by similar systemic factors—such as access to healthcare and health system outreach [111]. Additionally, the association can be understood through the lens of healthcare access and maternal engagement. When a woman is vaccinated with the tetanus toxoid containing vaccine during pregnancy, it shows that she has had contact with the healthcare system, which increases the chance that her child will also receive routine immunizations. Conversely, if a mother does not receive the tetanus vaccine, it may indicate broader barriers to healthcare, such as geographic isolation, lack of awareness, or systemic inequities.

Maternal education was negatively associated with zero-dose vaccination among children. Specifically, children of mothers with no formal education and those with only primary education had higher odds zero-dose vaccination, compared to children of mothers with secondary or higher education. This finding aligns with existing literature on the association between maternal education and vaccination uptake [34,39,112–114]. Usually, higher maternal educational status is associated with better health outcomes for children, including higher vaccination rates [112]. Educated mothers may have

greater access to health information, healthcare resources, and an understanding of the importance of childhood vaccination [113]. Therefore, engaging mothers with no or low education in community-based health initiatives or local vaccination campaigns that specifically target underserved populations is encouraged, as these programs often emphasize the importance of immunization and may be tailored to address the unique needs of communities with lower educational attainment [115].

Moreover, perceived distance to a health facility was significantly associated with zero-dose vaccination. Children from households where distance was a "big problem" had higher likelihood of zero-dose vaccination. This finding is in agreement with existing evidence [39,40]. The finding shows the impact of geographic accessibility on healthcare utilization. This could be explained by the following reasons: Firstly, the physical distance to healthcare facilities can be a barrier for many women, particularly those in rural or remote areas [42]. Long distances to a health facility often mean increased travel time and costs, which can avert mothers from seeking health care services for their children including vaccination. In addition, the perception of distance as a big problem can be influenced by the quality of transportation infrastructure. Poor road conditions, lack of reliable public transportation, and high transportation costs can exacerbate the challenges associated with accessing healthcare facilities [116–118]. It was also reported that people in rural areas often face long and difficult journeys to reach healthcare facility [117].

Finally, the odds of zero-dose vaccination were high among children living in rural areas when compared to children living in urban communities. Other studies in low- and middle-income countries have also reported similar finding [42,43,68,119–121].. This urban-rural disparity could be largely related systemic inequities in healthcare access and infrastructure. In sub-Saharan Africa, urban centers have a higher concentration of health facilities and personnel while rural areas often suffer from a chronic shortage of health facilities, trained personnel, and reliable immunization supply chains [122]. In addition, the WHO reported that between 2019 and 2022, over 28 million zero-dose children were recorded in the African Region, with the majority concentrated in countries where rural health systems are under-resourced [14].

## Strength and limitation

The strengths of this study were (1) the use of nationally representative large data from 28 SSA countries and (2) the application of multilevel spatial modeling, appropriate for hierarchical data that exhibit spatial dependence, as ignoring spatial dimension can lead to biased estimates and erroneous conclusions.

However, our study was not conducted without limitations. First, the reliance of our study on a secondary data source (DHS) restricted the inclusion of potentially important and unmeasured variables such as cultural beliefs, conflict, migration, or vaccine supply chain disruptions. Second, the estimated overall prevalence of zero-dose vaccination was with significant heterogeneity although we did subgroup analysis. Third, the immunization data are based on either maternal recall or vaccination cards, which introduces the possibility of recall or reporting bias. Moreover, while the study did identify spatial clustering and hot spot areas, the story might have been changed in countries with older DHS data, particularly among highly mobile populations. Finally, only two countries from the southern African region were included, which may limit the generalizability of the findings to that subregion. Therefore, the reader should take into account these limitations while interpreting the findings of this study.

## Conclusion

In sub-Saharan Africa, zero-dose vaccination among children aged 12–23 months was high when compared to the global target to leave no child behind. The distribution of zero-dose vaccination was spatially clustered, with the hot spot areas concentrated in most parts of Nigeria, Angola, Ethiopia, Madagascar, Guinea, Cote d'Ivoire, the southern border of Mauritania, and southeastern parts of Mali and Mozambique. The poorest household wealth index, home delivery, having no media exposure, no or low antenatal care visits, having no tetanus toxoid vaccination, no or primary maternal education,

perceived big problem of distance to health facilities and rural residence were significantly associated with zero-dose vaccination.

**Recommendation**

1. **Ministries of Health and National Governments in sub-Saharan Africa**
   - Implement spatially targeted immunization campaigns in identified hot spot regions
   - Expand access to education aimed at increasing literacy rate for the enhancement of health care service utilization including vaccination.

2. **International Organizations (Gavi, WHO, UNICEF, UNFPA)**
   - Provide technical and financial support for households with poorest wealth quintile in the hot spot areas of zero-dose vaccination in sub-Saharan Africa.
   - Support for ensuring availability and accessibility of health care facility among communities where distance to health facility is a big problem.

3. **Local Governments and Community Health Authorities**
   - Engage local leaders and influencers to promote institutional delivery and maternal health service utilization such as antenatal care and vaccination during pregnancy.

4. **Media and Communication Agencies**
   - Leveraging local radio, TV, and newspaper as media to create awareness about importance of childhood vaccination.

5. **Researchers**
   - Bayesian spatial hierarchical modeling of zero-dose vaccination among children to account heat mapping.

## Supporting information

**S1 File. Suplimentary Material.**
(DOCX)

**S2 File. Minimal Data.**
(ZIP)

## Author contributions

**Conceptualization:** Tadesse Tarik Tamir, Muluken Chanie Agimas, Dessie Abebaw Angaw.

**Data curation:** Tadesse Tarik Tamir, Muluken Chanie Agimas, Dessie Abebaw Angaw.

**Formal analysis:** Tadesse Tarik Tamir.

**Investigation:** Tadesse Tarik Tamir, Muluken Chanie Agimas, Dessie Abebaw Angaw.

**Methodology:** Tadesse Tarik Tamir, Muluken Chanie Agimas, Dessie Abebaw Angaw.

**Software:** Tadesse Tarik Tamir.

**Supervision:** Tadesse Tarik Tamir, Muluken Chanie Agimas, Dessie Abebaw Angaw.

**Validation:** Dessie Abebaw Angaw.

**Visualization:** Muluken Chanie Agimas, Dessie Abebaw Angaw.

**Writing – original draft:** Tadesse Tarik Tamir.

**Writing – review & editing:** Tadesse Tarik Tamir, Muluken Chanie Agimas, Dessie Abebaw Angaw.

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
