## [Decision Letter · Decision Letter 0]

6 Jan 2026

Dear Dr. Tamir,

Thank you for submitting your manuscript to PLOS ONE. After careful consideration, we feel that it has merit but does not fully meet PLOS ONE’s publication criteria as it currently stands. Therefore, we invite you to submit a revised version of the manuscript that addresses the points raised during the review process.

We look forward to receiving your revised manuscript.

Kind regards,

Edison Arwanire Mworozi, M.D

Academic Editor

PLOS One

“The authors did not receive a specific grant for this work from any funding agency in the public, private, or not-for-profit sectors.”

4. We note that Figures 1, 8, 9 and 10 in your submission contain map images which may be copyrighted. All PLOS content is published under the Creative Commons Attribution License (CC BY 4.0), which means that the manuscript, images, and Supporting Information files will be freely available online, and any third party is permitted to access, download, copy, distribute, and use these materials in any way, even commercially, with proper attribution. For these reasons, we cannot publish previously copyrighted maps or satellite images created using proprietary data, such as Google software (Google Maps, Street View, and Earth). For more information, see our copyright guidelines: http://journals.plos.org/plosone/s/licenses-and-copyright.

1. You may seek permission from the original copyright holder of Figure(s) [#] to publish the content specifically under the CC BY 4.0 license.

5. We are unable to open your Supporting Information file “Minimal Data.zip”. Please kindly revise as necessary and re-upload.

Additional Editor Comments:

Please address comments by reviewer one and ensure to resubmit a proper evidence backed paper in a lucid manner.

Reviewers' comments:

Reviewer's Responses to Questions

**Comments to the Author**

1. Is the manuscript technically sound, and do the data support the conclusions?

Reviewer #1: Yes

Reviewer #2: Yes

2. Has the statistical analysis been performed appropriately and rigorously?

Reviewer #1: Yes

Reviewer #2: Yes

3. Have the authors made all data underlying the findings in their manuscript fully available?

Reviewer #1: Yes

Reviewer #2: Yes

4. Is the manuscript presented in an intelligible fashion and written in standard English?

Reviewer #1: Yes

Reviewer #2: Yes

Reviewer #1: The purpose of this study is to conduct a mathematical analysis of the prevalence, spatial distribution, and determinants of zero-dose vaccination among children aged 12 to 23 months in sub-Saharan Africa applying a spatial multilevel modeling approach, thereby informing more targeted and effective immunization strategies. However, this analysis involves the use of complex mathematical models to investigate factors that have already been described before the current work using simpler methods. Moreover, in the discussion, the authors point out that the results they obtained are similar to the results obtained before this work. In the process of reading the materials of the article, a clinician may have difficulties interpreting and perceiving the presented material. In this regard, this scientific work is more suitable for specialists in the field of statistics, since it contains a large number of specific terms and approaches. It is necessary to shorten the description of the methods used and provide links to relevant works in which these approaches are described in detail.

The Introduction section does not indicate the advantage of the analysis method used, which would distinguish it from other informative methods. This may reduce the reader's interest in new data and results.

The results of the study are well described, but the section also needs to be shortened.

The Discussion section notes that the study revealed the prevalence, spatial variation, and individual- & community-level factors of zero-dose vaccination among children aged 12-23 months in sub-Saharan Africa using data from the 2015 to 2024 DHS with the application of a multilevel spatial model which accounts for the hierarchical nature of the data with its spatial dimension. However, it is further indicated that the obtained prevalence is consistent with the data previously reported in the systematic review of 204 countries: this indicator was almost the same, and the prevalence of zero-dose use ranged from 13.4% to 15.9%. A similar situation is observed for other results obtained for other factors under study. In this regard, the question arises about the expediency of conducting research and writing material. During the study, no new data was obtained or previously unknown factors and patterns were identified. In this regard, if new conclusions have been drawn in the course of the work, the authors should present them in a clearer form, comparing them with the results of other similar studies and demonstrating the advantages of the analysis method used. Otherwise, there is a risk that any statistical analysis method can be used to interpret data that is already known.

The material of the article seems to be voluminous and difficult to understand, therefore, in my opinion, it requires revision and reduction.

However, it is worth noting that the manuscript ends with recommendations for international and national authorities, as well as stakeholders, that can help minimize the number of zero-dose vaccinated children aged 12 to 23 months in sub-Saharan Africa.

Reviewer #2: The methodology is well explained, the discussion part is well elaborated and results explicitly identify the gaps which can be covered by the respective immunization programmes factoring in the hot spots

**Do you want your identity to be public for this peer review?** For information about this choice, including consent withdrawal, please see our Privacy Policy

Reviewer #1: **Yes:** Mikhail P. Kostinov

Reviewer #2: No

---

## [Author Response · Author response to Decision Letter 1]

21 Jan 2026

Subject: Submission of revised manuscript

Manuscript ID: [PONE-D-25-38190] - [EMID:8abf81ed5e5104bf]

Title: Prevalence, Spatial Patterns and Determinants of Zero-Dose Vaccination among Children Aged 12-23 Months in Sub-Saharan Africa: A Multilevel Spatial Analysis

Journal: PLOS One

I hope this letter finds you well. We appreciate the diligent efforts of the editorial team in facilitating the review process for our manuscript. Additionally, we extend our gratitude to the editors and reviewers for their valuable time and thoughtful feedback, which significantly contributed to enhancing the quality of our work.

The constructive comments provided by the reviewers have been instrumental in refining our study. We are pleased to note that the reviewers share our assessment of the scientific significance of our findings. In response to their suggestions, we have meticulously addressed each point raised. Please find our comprehensive responses to the comments below.

Furthermore, I have attached the revised manuscript file separately for your convenience. We believe that the revisions strengthen the manuscript and align it more closely with the journal’s scope and standards.

Thank you for considering our work for publication. We hope that our revised submission meets the high standards set by PLOS One.

Best regards,

Corresponding Author

Response to comments

Reviewer #1: The purpose of this study is to conduct a mathematical analysis of the prevalence, spatial distribution, and determinants of zero-dose vaccination among children aged 12 to 23 months in sub-Saharan Africa applying a spatial multilevel modeling approach, thereby informing more targeted and effective immunization strategies. However, this analysis involves the use of complex mathematical models to investigate factors that have already been described before the current work using simpler methods. Moreover, in the discussion, the authors point out that the results they obtained are similar to the results obtained before this work. In the process of reading the materials of the article, a clinician may have difficulties interpreting and perceiving the presented material. In this regard, this scientific work is more suitable for specialists in the field of statistics, since it contains a large number of specific terms and approaches. It is necessary to shorten the description of the methods used and provide links to relevant works in which these approaches are described in detail.

Response: Dear Reviewer, thank you for your thoughtful and detailed feedback. We appreciate your observation regarding the complexity of the methods section. Our initial intention was to provide sufficient detail to ensure transparency and reproducibility of the analysis. However, we fully understand your concern and have revised the manuscript accordingly. Specifically, we have shortened the methods section in the main text and moved the detailed analytical procedures to a supplementary file. Additionally, we have included references to relevant works that describe these approaches in detail. We hope these changes address your concern and make the manuscript more accessible to a broader audience.

The Introduction section does not indicate the advantage of the analysis method used, which would distinguish it from other informative methods. This may reduce the reader's interest in new data and results.

Response: Thank you for highlighting this important point. We have substantially revised the introduction to clearly articulate the advantages of the spatial multilevel modeling approach and how it adds value beyond previous studies. The revised section emphasizes the novelty of incorporating spatial dimensions and hierarchical structures, which allows for more precise identification of geographic disparities and contextual factors influencing zero-dose vaccination. We believe this revision strengthens the rationale for our methodological choice.

The results of the study are well described, but the section also needs to be shortened.

Response: We appreciate your suggestion. In response, we have carefully revised and condensed the results section while retaining all essential findings. We believe the revised version is more concise and easier to follow. Please refer to the updated manuscript for these changes.

The Discussion section notes that the study revealed the prevalence, spatial variation, and individual- & community-level factors of zero-dose vaccination among children aged 12-23 months in sub-Saharan Africa using data from the 2015 to 2024 DHS with the application of a multilevel spatial model which accounts for the hierarchical nature of the data with its spatial dimension. However, it is further indicated that the obtained prevalence is consistent with the data previously reported in the systematic review of 204 countries: this indicator was almost the same, and the prevalence of zero-dose use ranged from 13.4% to 15.9%.

Response: Thank you for your constructive observation. While our pooled prevalence estimates are consistent with global systematic review findings, our study provides updated evidence specific to sub-Saharan Africa and highlights significant country-level variations. Several countries exhibit prevalence rates far above the global pooled estimate, which underscores persistent inequities in immunization coverage. This updated regional and country-specific evidence is critical for targeted interventions. We have clarified these points in the discussion to better demonstrate the added value of our analysis.

A similar situation is observed for other results obtained for other factors under study. In this regard, the question arises about the expediency of conducting research and writing material. During the study, no new data was obtained or previously unknown factors and patterns were identified. In this regard, if new conclusions have been drawn in the course of the work, the authors should present them in a clearer form, comparing them with the results of other similar studies and demonstrating the advantages of the analysis method used. Otherwise, there is a risk that any statistical analysis method can be used to interpret data that is already known.

Response: Thank you for raising this important concern. While many associations align with previous findings, our study provides new insights through advanced modeling. For instance, factors previously reported as significant (e.g., multiple wealth quintiles) were not significant in our analysis, and the strength of associations differed. Specifically, only the poorest wealth quintile remained significant, which contrasts with earlier studies. These nuanced findings, along with spatial clustering patterns, demonstrate the added value of our approach. We have revised the discussion to clearly highlight these new insights and compare them with prior studies.

The material of the article seems to be voluminous and difficult to understand, therefore, in my opinion, it requires revision and reduction.

Response: We appreciate your feedback and have made several revisions to improve clarity and reduce length. The methods and results sections have been streamlined, and detailed technical content has been moved to supplementary materials. We hope these changes make the manuscript more concise and reader-friendly.

However, it is worth noting that the manuscript ends with recommendations for international and national authorities, as well as stakeholders, that can help minimize the number of zero-dose vaccinated children aged 12 to 23 months in sub-Saharan Africa.

Reponse: Thank you for acknowledging the practical implications of our study. We appreciate your recognition of the recommendations provided and hope they contribute to actionable strategies for improving immunization coverage.

Reviewer #2: The methodology is well explained, the discussion part is well elaborated and results explicitly identify the gaps which can be covered by the respective immunization programmes factoring in the hot spots.

Response: Dear Reviewer, thank you for your positive assessment of our manuscript. We are pleased that you found the methodology, discussion, and identification of gaps to be clear and useful. Your encouraging feedback is greatly appreciated.

---

## [Decision Letter · Decision Letter 1]

19 Feb 2026

Prevalence, Spatial Patterns and Determinants of Zero-Dose Vaccination among Children Aged 12-23 Months in Sub-Saharan Africa: A Multilevel Spatial Analysis

PONE-D-25-38190R1

Dear authors,

We’re pleased to inform you that your manuscript has been judged scientifically suitable for publication and will be formally accepted for publication once it meets all outstanding technical requirements.

Kind regards,

Edison Arwanire Mworozi, M.D

Academic Editor

PLOS One

Additional Editor Comments (optional):

The paper is wonderful and ready to publish as per the reviewers comments!

Reviewers' comments:

Reviewer's Responses to Questions

**Comments to the Author**

Reviewer #1: All comments have been addressed

Reviewer #2: All comments have been addressed

2. Is the manuscript technically sound, and do the data support the conclusions?

Reviewer #1: Yes

Reviewer #2: Yes

3. Has the statistical analysis been performed appropriately and rigorously?

Reviewer #1: Yes

Reviewer #2: Yes

4. Have the authors made all data underlying the findings in their manuscript fully available?

Reviewer #1: Yes

Reviewer #2: Yes

5. Is the manuscript presented in an intelligible fashion and written in standard English?

Reviewer #1: Yes

Reviewer #2: Yes

Reviewer #1: The authors have taken into account all the previous comments and suggestions I made. There are no new comments, so I recommend the manuscript for publication.

Reviewer #2: The questions and comments of reviewers on previous iteration have been addressed to maximum extent.

**Do you want your identity to be public for this peer review?** For information about this choice, including consent withdrawal, please see our Privacy Policy

Reviewer #1: **Yes:** Mikhail P. Kostinov

Reviewer #2: **Yes:** Soofia Yunus

---

## [Editor Report · Acceptance letter]

PONE-D-25-38190R1

PLOS One

Dear Dr. Tamir,

I'm pleased to inform you that your manuscript has been deemed suitable for publication in PLOS One. Congratulations! Your manuscript is now being handed over to our production team.

Kind regards,

on behalf of

Professor Edison Arwanire Mworozi

Academic Editor

PLOS One